# Integrated culturing, modeling and transcriptomics uncovers complex interactions and emergent behavior in a three-species synthetic gut community

Kevin D'hoe[1,2,3,4‡], Stefan Vet[5,6,7‡], Karoline Faust[1‡*], Frédéric Moens[4], Gwen Falony[1,2], Didier Gonze[6,7], Verónica Lloréns-Rico[1,2], Lendert Gelens[8], Jan Danckaert[5], Luc De Vuyst[4†], Jeroen Raes[1,2,3†*]

[1]Laboratory of Molecular Bacteriology, KU Leuven Department of Microbiology and Immunology, Rega Institute, Leuven, Belgium; [2]Jeroen Raes Lab, VIB-KU Leuven Center for Microbiology, Leuven, Belgium; [3]Research Group of Microbiology, Department of Bioengineering Sciences, Vrije Universiteit Brussel, Brussels, Belgium; [4]Research Group of Industrial Microbiology and Food Biotechnology, Faculty of Sciences and Bioengineering Sciences, Vrije Universiteit Brussel, Brussels, Belgium; [5]Applied Physics Research Group, Vrije Universiteit Brussel, Brussels, Belgium; [6]Unité de Chronobiologie Théorique, Université Libre de Bruxelles, Brussels, Belgium; [7]Interuniversity Institute of Bioinformatics in Brussels, Brussels, Belgium; [8]Laboratory of Dynamics in Biological Systems, KU Leuven, Leuven, Belgium

*For correspondence:
karoline.faust@kuleuven.be (KF);
jeroen.raes@kuleuven.vib.be (JR)

†These authors also contributed equally to this work
‡These authors also contributed equally to this work

Competing interests: The authors declare that no competing interests exist.

**Abstract** The composition of the human gut microbiome is well resolved, but predictive understanding of its dynamics is still lacking. Here, we followed a bottom-up strategy to explore human gut community dynamics: we established a synthetic community composed of three representative human gut isolates (*Roseburia intestinalis* L1-82, *Faecalibacterium prausnitzii* A2-165 and *Blautia hydrogenotrophica* S5a33) and explored their interactions under well-controlled conditions in vitro. Systematic mono- and pair-wise fermentation experiments confirmed competition for fructose and cross-feeding of formate. We quantified with a mechanistic model how well tri-culture dynamics was predicted from mono-culture data. With the model as reference, we demonstrated that strains grown in co-culture behaved differently than those in mono-culture and confirmed their altered behavior at the transcriptional level. In addition, we showed with replicate tri-cultures and simulations that dominance in tri-culture sensitively depends on the initial conditions. Our work has important implications for gut microbial community modeling as well as for ecological interaction detection from batch cultures.
DOI: https://doi.org/10.7554/eLife.37090.001

## Introduction

The human gut microbiome is a complex, spatially heterogeneous and dynamic ecosystem consisting of hundreds of species interacting with each other and with the human host. It is a daunting task to develop predictive models for such a system, yet the potential rewards are high and would, for instance, enable targeted interventions to shift dysbiotic communities towards more healthy states. Two conditions need to be fulfilled for predictive models to be successful: first, the system has to be sufficiently well characterized to build the model; and, second, the dynamics should be generally

**eLife digest** Our gut is home to trillions of microorganisms, most of them bacteria, which have an important impact on our body. During healthy periods, these microorganisms help our digestion, protect our cells, and compete against disease-causing bacteria. But specific communities of gut bacteria are linked to many diseases.

We already have a good knowledge of the bacterial composition present in a wide range of human guts, but how the different bacterial species within such communities affect each other, has so far been unclear. Future disease treatments may be able to steer 'bad' communities to healthier mixtures. For this to happen we need to know how species interact and how these interactions change the behavior of the whole community.

To investigate this further, D'hoe, Vet, Faust et al. studied three common species of gut bacteria under controlled conditions in the laboratory. The different species were either grown alone, in pairs or together, and the number of bacteria and the concentration of nutrients were measured over time. The results showed that when grown alone or together, their behavior changed.

D'hoe et al. then used a mathematical model to estimate the rates at which species multiplied and consumed nutrients. This model was able to predict the dynamics of each of the species grown alone. However, the data from bacteria grown in pairs was needed to predict the dynamics of bacteria grown as a group of three. Next, D'hoe et al. compared the activity of genes between bacteria grown alone or together, and discovered several differences.

This suggests that bacterial species affect each other greatly, and community behavior cannot be predicted from knowledge of its members alone. Therefore, studying bacteria in isolation is not enough to understand the complex environments of our guts, which are inhabited not by three but hundreds of bacterial species. In future, interactions between bacteria will need to be studied to ultimately be able to shift the gut community into better shapes.

DOI: https://doi.org/10.7554/eLife.37090.002

deterministic. First successes in modeling the behavior of gut microbial communities give reason for cautious hope (*Buffie et al., 2015*; *Cremer et al., 2017*; *Muñoz-Tamayo et al., 2016*; *Stein et al., 2013*). Most of these studies took a top-down approach, in which the change in composition of an entire community in vivo is modeled. For instance, *Cremer et al. (2017)* predicted the ratio of Firmicutes and Bacteroidetes in fecal samples as a function of estimated water content and nutrient influx using a diffusion model. Others have fitted population models to time series of taxon (mostly genus) abundances obtained from 16S rRNA gene sequencing. For instance, one study fitted a variant of the generalized Lotka-Volterra (gLV) model to a cecal gut time series of mice exposed to the pathogen *Clostridium difficile*, an antibiotic or both, thereby inferring the interactions between different genera (*Stein et al., 2013*). The same approach was also used to predict species that inhibit *C. difficile* growth in murine and human microbiota, one of which significantly lowered mortality when transferred to mice before infection with *C. difficile* (*Buffie et al., 2015*).

Despite these successes, the gLV model and its variants have several drawbacks that limit their widespread application. gLV-type models describe species dynamics as a function of their growth rates and pairwise interactions, without taking the concentrations of exchanged metabolites into account. Thus, they assume that community dynamics can be predicted from pair-wise interactions and that the interaction mechanisms can be ignored. These assumptions have recently been tested both experimentally and computationally: *Friedman et al. (2017)* experimentally quantified the accuracy reached when predicting the behavior of more complex soil communities from species pairs, whereas *Momeni et al. (2017)* systematically compared LV models of metabolite-mediated species interactions to their mechanistic counterparts. While the authors in the former case concluded that the behavior of larger communities could, to a considerable extent, be predicted from that of smaller ones, the latter study showed that the (extended) gLV model cannot accurately describe several common types of interaction mechanisms.

An alternative to the gLV model and its variants are mechanistic models, which in contrast to gLV models account for metabolite-mediated interactions by explicitly describing the dynamics of the produced and consumed compounds (see *Momeni et al., 2017* and references therein). They thus

require more system knowledge than generic gLV and related models do. However, most members of the gut community have not been thoroughly characterized, and little is known about their responses to different nutrients, pH values and interaction partners, even for those that have been studied more closely. It is challenging to obtain this type of biological knowledge and to resolve interaction mechanisms in vivo. However, in vitro studies allow the acquisition of detailed knowledge not only of the microorganisms' pH and nutrient preferences but also of their behavior in the presence of other microorganisms. In vitro studies of human gut microorganisms have a long tradition and have been carried out in several different ways. Classical mono- and co-culture studies in batch and chemostat fermentors have explored nutrient preferences and interaction mechanisms (*Falony et al., 2006*; *Falony et al., 2009a*; *Moens et al., 2016*; *Moens et al., 2017*; *Rivière et al., 2016*). Artificial gut systems, such as the TNO In Vitro Model of the Colon (TIM-2) (*Venema, 2015*) and the Simulation of the Human Intestinal Microbial Ecosystem (SHIME) (*Van de Wiele et al., 2015*), seek to reproduce the conditions of the human gastro-intestinal tract as closely as possible and in a well-controlled manner. The gut community has also been studied in vitro at smaller scales, in minibioreactor arrays (*Auchtung et al., 2015*) and with gut-on-chip microfluidic devices (*Kim et al., 2012*; *Shah et al., 2016*).

In most cases, however, gut simulators are inoculated with fecal material. In the range from top-down to bottom-up approaches that explore gut microbial community dynamics, these can be considered as intermediate cases, in which the host is eliminated but the community is not further simplified. The goal of these studies is usually to quantify the behavior of the entire community under different conditions. In the cases of HuMiX and of SHIME's HMI module, the interaction of particular gut microorganisms with epithelial cells is targeted (*Marzorati et al., 2014*; *Shah et al., 2016*). As the exact composition of fecal material (which also includes bacteriophages and fungi) is difficult to resolve, it is hard to track each member in such a community. Although the in vitro dynamics of colon (*Kettle et al., 2015*) and rumen (*Muñoz-Tamayo et al., 2016*) communities has been described with mechanistic models previously, these models did not account for the behavior at species level, and instead grouped species with similar metabolic activities into guilds. While it is of interest to model guild dynamics, the resolution of guild-level models may be insufficient to provide an understanding of microbial community dynamics in the gut. Species in the same guild do not necessarily respond in the same manner to altered environments and perturbations. Guild definitions are arbitrary to an extent, and gut bacteria with flexible metabolic strategies may change their guild membership. In addition, the concepts of tipping elements (*Lahti et al., 2014*) and strongly interacting species (*Gibson et al., 2016*) suggest that particular species can have a disproportionate impact on gut community dynamics.

In our opinion, experiments using defined communities of known composition, grown under well-controlled conditions, are crucial to learn more about the interactions of gut species and how these shape community dynamics. Well-controlled in vitro experiments are also necessary for the development and validation of predictive models of gut microbial communities. However, only a few in vitro experiments with defined gut communities have been reported to date (*Newton et al., 2013*; *Trosvik et al., 2008*; *Trosvik et al., 2010*) and only one study has, to our knowledge, employed mechanistic models to predict community dynamics in the infant gut microbiome (*Pinto et al., 2017*). The objective of the present study was therefore to establish a defined community composed of human strains that are representative of the adult gut microbiome, to study their interactions under well-controlled conditions in vitro and to validate a quantitative mechanistic model by predicting community behavior in a tri-culture with parameters from mono-culture data. Mechanistic models have been tested in this manner before for a cystic fibrosis community (*Schmidt et al., 2011*) but such an approach has not yet been applied to a synthetic gut community.

To reach our objective, we created a synthetic community composed of three abundant and typical members of the human gut microbiome: *Faecalibacterium prausnitzii* A2-165 (*Duncan et al., 2002b*), *Roseburia intestinalis* L1-82 (*Duncan et al., 2002a*) and *Blautia hydrogenotrophica* S5a33 (*Bernalier et al., 1996*). All three strains were isolated from human feces and their draft genomes are available. Furthermore, they are of particular medical relevance because of the ability of two of these strains (*R. intestinalis* L1-82 and *F. prausnitzii* A2-165) to produce butyrate, a beneficial short chain fatty acid that is an important energy source for gut epithelial cells (*Geirnaert et al., 2017*; *Rivière et al., 2016*). Butyrate producers are often depleted in dysbiotic gut microbiota relative to

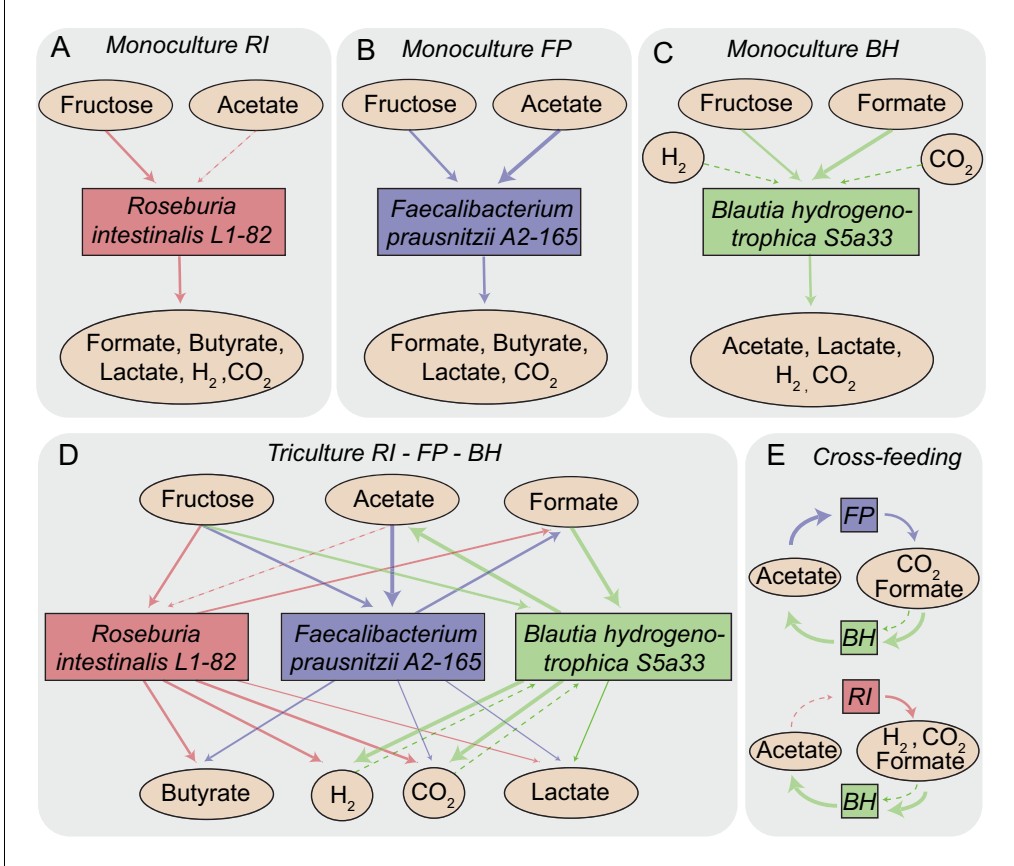

**Figure 1.** Overview of metabolite-mediated strain interactions. (**A–C**) Strain-specific metabolite consumption and production. (**D**) Metabolite-mediated interactions present in the tri-culture. (**E**) Cross-feeding interactions between *Faecalibacterium prausnitzii* A2-165 (FP) and *Blautia hydrogenotrophica* S5a33 (BH) as well as between *Roseburia intestinalis* L1-82 (RI) and BH. The dashed arrow from acetate to RI denotes net acetate consumption. The dashed arrows from hydrogen and $CO_2$ to BH indicate the potential of this bacterium to grow autotrophically on these gasses.

DOI: https://doi.org/10.7554/eLife.37090.003

healthy controls (*Antharam et al., 2013*; *Rivera-Chávez et al., 2016*). Thus, high butyrate production will probably be a quality criterion for bacterial cocktails designed for therapeutic purposes.

In *R. intestinalis* L1-82, fermentation of carbohydrates results in the production of butyrate as well as hydrogen gas and carbon dioxide (*Duncan et al., 2002a*; *Falony et al., 2009c*), whereas *F. prausnitzii* A2-165 produces formate in addition to butyrate and requires acetate for growth (*Duncan et al., 2002b*; *Moens et al., 2016*). *B. hydrogenotrophica* S5a33 is able to grow on carbon dioxide and hydrogen gas, but also on glucose and fructose, in all cases generating acetate (*Bernalier et al., 1996*). Therefore, as *Figure 1* illustrates, our community contains multiple cross-feeding and competitive interactions. For instance, all three strains compete for fructose. *B. hydrogenotrophica* S5a33 can use the hydrogen gas generated by *R. intestinalis* L1-82 as well as the carbon dioxide and formate produced by both *R. intestinalis* L1-82 and *F. prausnitzii* A2-165. In turn, *B. hydrogenotrophica* S5a33 provides acetate that is beneficial to *R. intestinalis* L1-82 and *F. prausnitzii* A2-165. This system thus constitutes a rare example of two strain pairs that simultaneously compete and mutually cross-feed.

The three strains were grown as mono-, bi-, or tri-cultures in 2 L laboratory fermentors in batch mode. We monitored the dynamics of each combination by quantifying bacteria through optical density (OD), flow cytometry and qPCR and by measuring the concentration of substrates and fermentation products, including short chain fatty acids and gasses. Finally, we sequenced the total RNA in selected samples. *Figure 2* summarizes our approach. To our knowledge, this is the first study to investigate a synthetic gut community with a combination of mono- and co-cultures, mechanistic modeling and gene expression analysis.

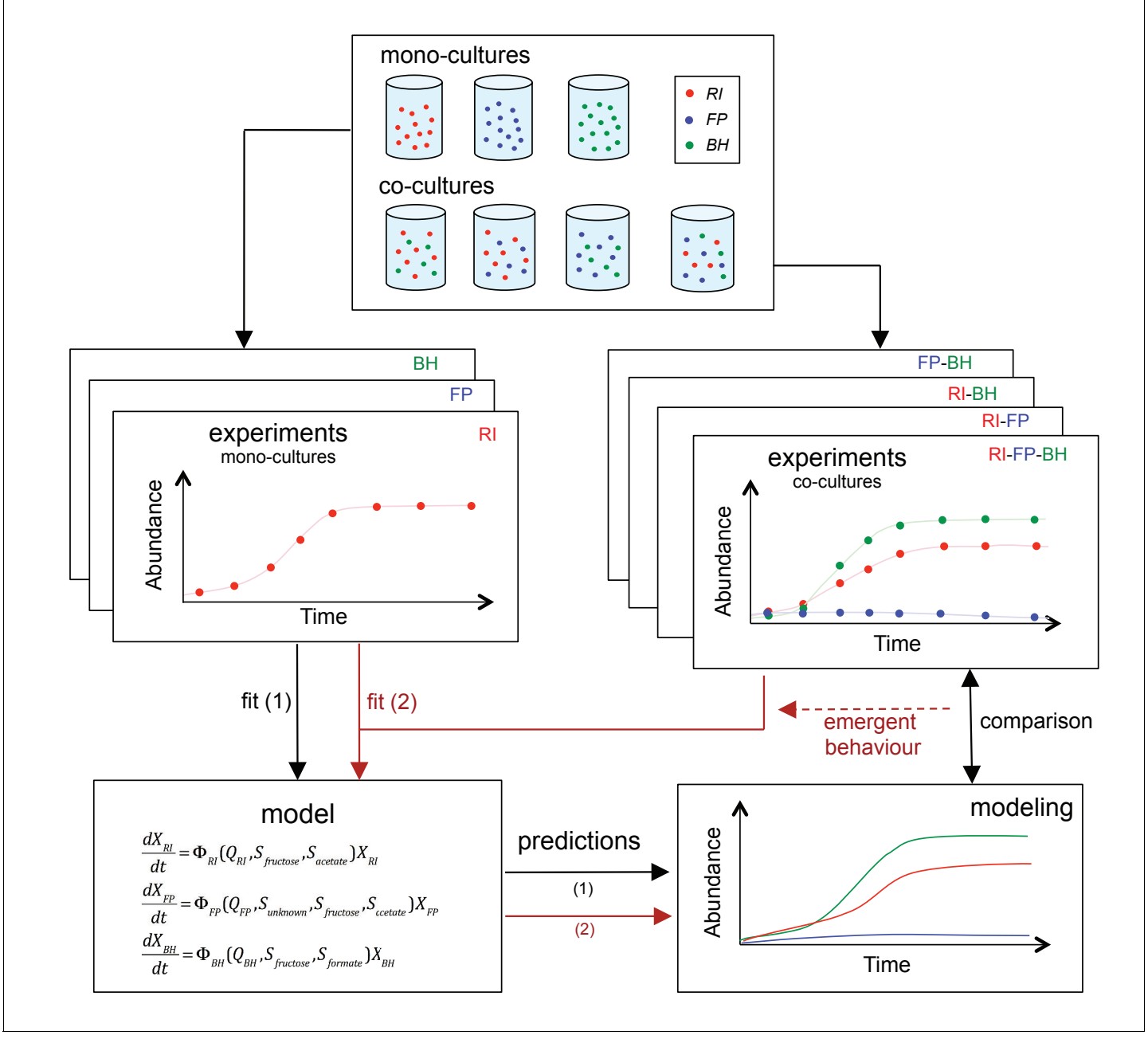

**Figure 2.** Scheme summarizing the experimental set-up and modeling approach. A mechanistic model of a three-strain community consisting of *Roseburia intestinalis* L1-82 (RI), *Faecalibacterium prausnitzii* A2-165 (FP) and *Blautia hydrogenotrophica* S5a33 (BH) is parameterized on mono-cultures, but does not describe tri-culture dynamics well. Data from bi-cultures are taken into account to improve the goodness of fit to the tri-culture data, thereby indicating emergent behavior.

DOI: https://doi.org/10.7554/eLife.37090.004

## Results

### *Blautia hydrogenotrophica* S5a33 consumes fructose and formate

We first confirmed the cross-feeding interactions postulated for *B. hydrogenotrophica* S5a33 with small-volume screening experiments, in which the pH was not kept constant and the atmosphere contained 10% carbon dioxide and 10% hydrogen gas. We found that under these conditions, *B. hydrogenotrophica* S5a33 was able to grow heterotrophically on formate, which was entirely

**Table 1.** Overview of fermentation experiments and model fitting results (RMSE: root mean square error). Means and standard deviations are reported across biological replicates.

| Strains | Number of biological replicates | Added energy source and/or co-substrate | Consumption (-)/production (+) of metabolites in mM | | | | | | | Carbon recovery (%) | O/R balance | Selected for parameterization 1 | RMSE parameterization 1 | Selected for parameterization 2 | RMSE parameterization 2 |
|---|---|---|---|---|---|---|---|---|---|---|---|---|---|---|---|
| | | | Fructose | Acetate | Butyrate | Formate | Lactate | $H_2$ | $CO_2$ | | | | | | |
| **Mono culture fermentations** | | | | | | | | | | | | | | | |
| R. intestinalis L1-82 | 4 | Fructose/acetate | −48.8 ± 0.9 | −20.6 ± 2.4 | 58.5 ± 3.9 | 6.8 ± 1.2 | 3.9 ± 1.1 | 72.2 ± 4.9 | 84.0 ± 0.8 | 105 ± 14.3 | 0.86 ± 0.2 | Yes | 0.17 ± 0.04 | No | 0.83 ± 0.13 |
| F. prausnitzii A2-165 | 3 | Fructose/acetate | −23.1 ± 1.1 | −13.5 ± 0.8 | 29.8 ± 0.2 | 22.7 ± 0.9 | 1.8 ± 0.2 | 0.2 ± 0.2 | 19.3 ± 1.6 | 100.9 ± 3.4 | 1.05 ± 0.09 | Yes | 0.15 ± 0.08 | Yes | 0.15 ± 0.08 |
| B. hydrogenotrophica S5a33 | 3 | Fructose/formate | −19.0 ± 5.6 | 23.0 ± 6.6 | 0.0 ± 0.0 | −36.1 ± 1.8 | 6.0 ± 1.2 | 31.5 ± 7.7 | 26.0 ± 5.3 | 60.0 ± 2.1 | 0.76 ± 0.07 | Yes | 1.43 ± 0.27 | No | 3.53 ± 1.04 |
| **Bi-culture fermentations** | | | | | | | | | | | | | | | |
| R. intestinalis L1−82/F. prausnitzii A2-165 | 2 | Fructose/acetate | −47.7 ± 2.2 | −18.1 ± 2.7 | 56.3 ± 4.5 | 7.6 ± 2.7 | 4.3 ± 3.3 | 114 ± 16 | 85.7 ± 4.1 | 102.8 ± 0.3 | 0.79 ± 0 | No | 0.77 ± 0.06 | No | 0.68 ± 0.03 |
| R. intestinalis L1−82/B. hydrogenotrophica S5a33 | 2 | Fructose/acetate | −46.5 ± 1.5 | −7.5 ± 1.9 | 53.5 ± 4.7 | −0.5 ± 0.8 | 2.2 ± 0.3 | 53.0 ± 3.0 | 74.9 ± 4.1 | 100.2 ± 0.6 | 0.94 ± 0.12 | No | 0.78 ± 0.36 | Yes | 0.3 ± 0.04 |
| R. intestinalis L1−82/B. hydrogenotrophica S5a33 | 2 | Fructose | −48.0 ± 0.2 | 44.9 ± 5.4 | 27.9 ± 3.0 | −1.0 ± 0.0 | 5.4 ± 2.4 | 33.1 ± 3.2 | 47.4 ± 3.9 | 91.6 ± 1.1 | 1.05 ± 0.06 | No | 0.9 ± 0.17 | No | 0.32 ± 0.04 |
| F. prausnitzii A2−165/B. hydrogenotrophica S5a33 | 2 | Fructose/acetate | −49.3 ± 1.0 | 41.4 ± 11.7 | 30.7 ± 1.8 | −1.2 ± 0.0 | 3.9 ± 1.6 | 56.2 ± 37.1 | 54.8 ± 33.3 | 91.6 ± 9.4 | 0.88 ± 0.25 | No | 0.6 ± 0.14 | No | 0.26 ± 0.13 |

*Table 1 continued on next page*

*Table 1 continued*

| Strains | Number of biological replicates | Added energy source and/or co-substrate | Consumption (-)/production (+) of metabolites in mM | | | | | | | Carbon recovery (%) | O/R balance | Selected for parameterization 1 | RMSE parameterization 1 | Selected for parameterization 2 | RMSE parameterization 2 |
|---|---|---|---|---|---|---|---|---|---|---|---|---|---|---|---|
| | | | Fructose | Acetate | Butyrate | Formate | Lactate | $H_2$ | $CO_2$ | | | | | | |
| *F. prausnitzii A2–165/B. hydrogenotrophica S5a33* | 1 | Fructose | −47.0 | 62.5 | 25.5 | −1.1 | 4.5 | 62.9 | 63.6 | 107.4 | 1.11 | No | 0.63 | No | 0.46 |
| Tri-culture fermentations | | | | | | | | | | | | | | | |
| *R. intestinalis L1–82/F. prausnitzii A2–165/B. hydrogenotrophica S5a33* | 6 | Fructose | −48.9 ± 1.3 | 38.4 ± 16.9 | 32.5 ± 5.5 | −1.2 ± 0.1 | 7.4 ± 1.7 | 61.9 ± 3.2 | 57.1 ± 9.2 | 97.0 ± 3.1 | 0.84 ± 0.09 | No | 0.78 ± 0.27 | No | 0.58 ± 0.11 |

DOI: https://doi.org/10.7554/eLife.37090.005

consumed. Although we did not quantify gasses during screening and therefore could not ascertain the consumption of carbon dixoide and hydrogen gas, we observed growth in the absence of an added carbon source, indicating autotrophic growth as described previously (*Bernalier et al., 1996*). Presumably, both formate and carbon dioxide are assimilated via the Wood-Ljungdahl pathway, of which all required genes are present in the genome of *B. hydrogenotrophica* S5a33 according to the AGORA database (*Magnúsdóttir et al., 2017*).

We also found that *B. hydrogenotrophica* S5a33 grew on fructose, oligofructose and glucose, as reported by *Rey et al. (2010)* for *B. hydrogenotrophica* S5a36, and documented partial consumption of these saccharides. For glucose and fructose, the maximal OD tended to be lower than for

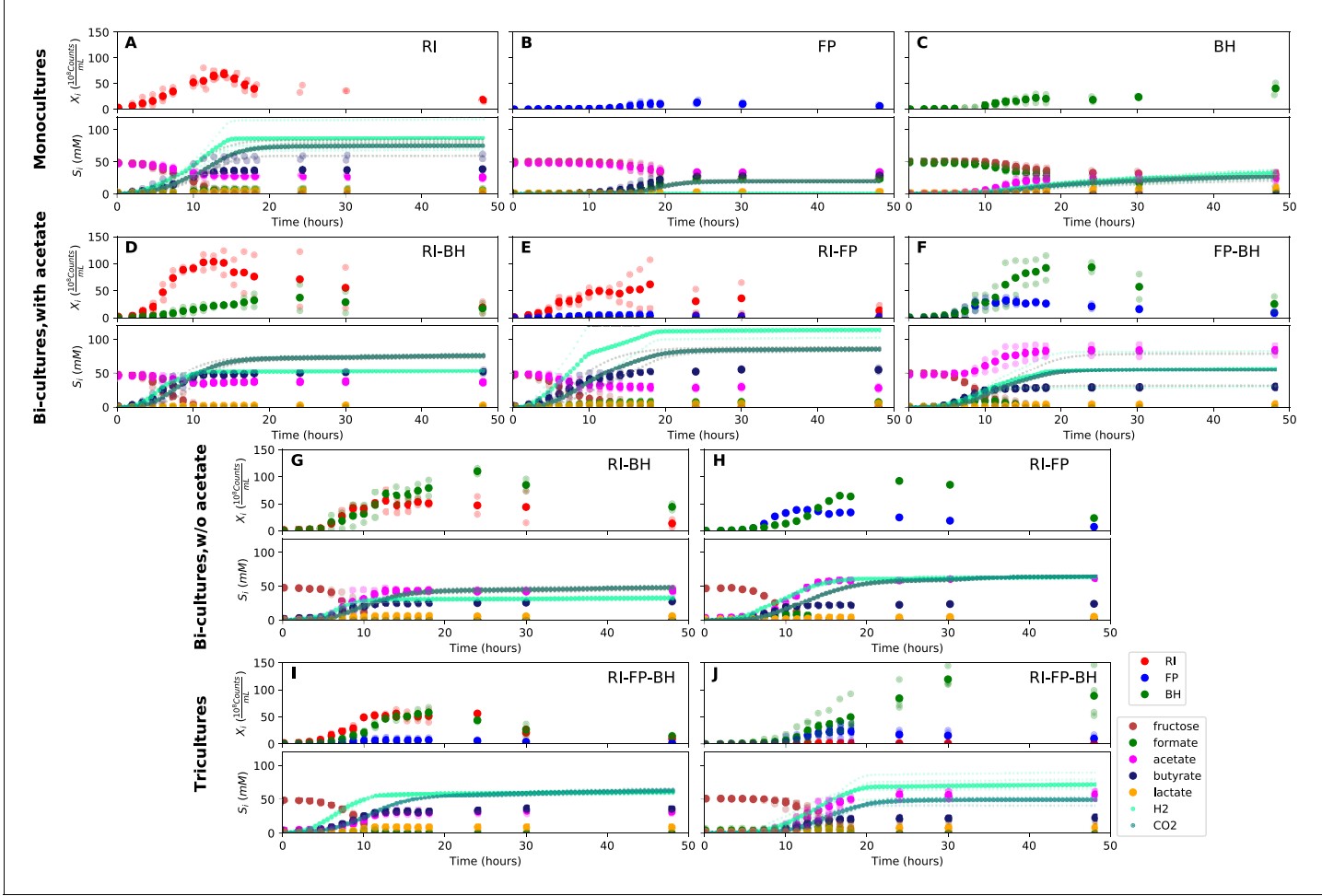

**Figure 3.** Summary of fermentation data. Biological replicates are plotted together in one panel, with their mean shown in bold. For each set of experiments, species abundances quantified by qPCR are plotted in the top half of the panel and metabolite concentrations in the bottom half. (**A–C**) Monocultures of *Roseburia intestinalis* L1-82 (RI), *Faecalibacterium prausnitzii* A2-165 (FP) and *Blautia hydrogenotrophica* S5a33 (BH). (**D–F**) The three co-culture combinations of RI, FP and BH with initial acetate. (**G–H**) Co-cultures of RI versus BH and FP versus BH without initial acetate. (**I–J**) The tri-culture replicates are separated into those dominated by RI and BH (**I**) and those dominated by FP and BH (**J**).

DOI: https://doi.org/10.7554/eLife.37090.006

The following source data and figure supplements are available for figure 3:

**Source data 1.** The qPCR data and HPLC measurements are reported as the mean across three technical replicates for each of the fermentation experiments shown in *Figure 3*.
DOI: https://doi.org/10.7554/eLife.37090.009

**Figure supplement 1.** Test for prokaryotic contamination with 16S rRNA gene sequencing.
DOI: https://doi.org/10.7554/eLife.37090.007

**Figure supplement 2.** Test for viral, prokaryotic and eukaryotic contamination in RNA-seq data.
DOI: https://doi.org/10.7554/eLife.37090.008

formate (mean maximal ODs for glucose: 0.3, fructose: 0.6, formate: 1.3). In agreement with (*Bernalier et al., 1996*), we detected lactate in addition to acetate for these substrates and confirmed lactate production in the presence of fructose in the fermentor. Notably, when growing *B. hydrogenotrophica* S5a33 on formate but without fructose in the fermentor, carbon dioxide and hydrogen gas were produced besides acetate, but lactate was absent. *B. hydrogenotrophica* S5a33 also consumed small concentrations of galactose, but did not consume fucose, inulin or lactate. In conclusion, we confirmed the potential competition between *B. hydrogenotrophica* S5a33 and the two primary fermenters for fructose as well as the potential cross-feeding of formate.

## Mono-culture dynamics does not follow standard Monod kinetics

We employed pH-controlled mono-cultures to characterize the properties and growth kinetics of the individual strains in our model. *Table 1* provides an overview of all of the fermentation experiments carried out, whereas *Supplementary file 1* gives additional information for each experiment.

When grown in monoculture, *R. intestinalis* L1-82 consumed fructose and produced butyrate, carbon dioxide and hydrogen gas, as described previously (*Falony et al., 2009c*), as well as small amounts of lactate and formate (*Figure 3A*). Interestingly, there was no net consumption of acetate when more fructose than acetate was provided. Net acetate consumption has been found to correlate negatively with hydrogen gas production (*Falony et al., 2009c*), but here we saw that it also depended on the ratio of initial fructose and acetate. When given in equal concentrations, *R. intestinalis* L1-82 partially consumed acetate. Consequently, in all further experiments, when acetate was added, it was added at the same concentration as fructose.

*F. prausnitzii* A2-165 in monoculture produced formate, less carbon dioxide and butyrate than *R. intestinalis* L1-82 and no hydrogen gas, but did not entirely consume fructose (*Figure 3B*). After having excluded a number of explanations — exposure to oxygen (by adding oxygen gas via sterile water), redox potential (by continuously adding the oxidizing agent potassium ferrocyanide trihydrate), pH (lowered to 5.8), a threshold requirement for fructose (halving the fructose concentration did not stop its consumption) or end-product inhibition (by adding initial butyrate) — we found that doubling the concentration of yeast extract lowered residual fructose concentrations. Adding fresh but autoclaved medium during the fermentation did not lower residual fructose concentrations, so we assumed that *F. prausnitzii* A2-165 was growth-limited by one or several heat-labile co-factor(s) present in the yeast extract. A recent flux balance analysis with a manually curated metabolic reconstruction suggests that the growth of *F. prausnitzii* A2-165 requires several amino acids (L-alanine, L-cysteine, L-methionine, L-serine and L-tryptophan) and the co-factors biotin (vitamin $B_7$), cobalamin (vitamin $B_{12}$), folic acid (vitamin $B_9$), hemin, nicotinic acid, pantothenic acid and riboflavin (vitamin $B_2$) (*Heinken et al., 2014*). With the exceptions of cobalamin and externally supplied hemin, these nutrients should be present in yeast extract according to the metabolic reconstruction of *Saccharomyces cerevisiae* iMM904 (*Mo et al., 2009*) and, furthermore, the amino acids should be present in other medium components (bacteriological peptone, soy peptone and tryptone). According to previous experimental findings as well as the flux balance analysis, *F. prausnitzii* A2-165 can grow in the presence of oxygen gas (*Heinken et al., 2014*; *Khan et al., 2012*), which is in agreement with our observation that the addition of low concentrations of oxygen gas does not alter its growth curve. *F. prausnitzii* A2-165 is assumed to transfer electrons to oxygen through extracellular redox mediators such as riboflavin (*Khan et al., 2012*; *Prévoteau et al., 2015*).

*B. hydrogenotrophica* S5a33 produced acetate, hydrogen gas, carbon dioxide and small concentrations of lactate, while consuming formate almost entirely (*Figure 3C*). It also consumed fructose, but did not deplete it. While the carbon recovery for *F. prausnitzii* A2-165 and *R. intestinalis* L1-82 monocultures was close to 100%, it only reached 60% for *B. hydrogenotrophica* S5a33 in monoculture on formate and fructose.

These unexpected behaviors defy simple kinetic models typically based on additive Monod functions and necessitate adjustment of the equations.

## Prediction accuracy of the model parameterized on monocultures is strain-dependent

We designed a model that described the dynamics of each strain and of key compounds (including fructose, formate, acetate, butyrate, hydrogen gas and carbon dioxide) with ordinary differential

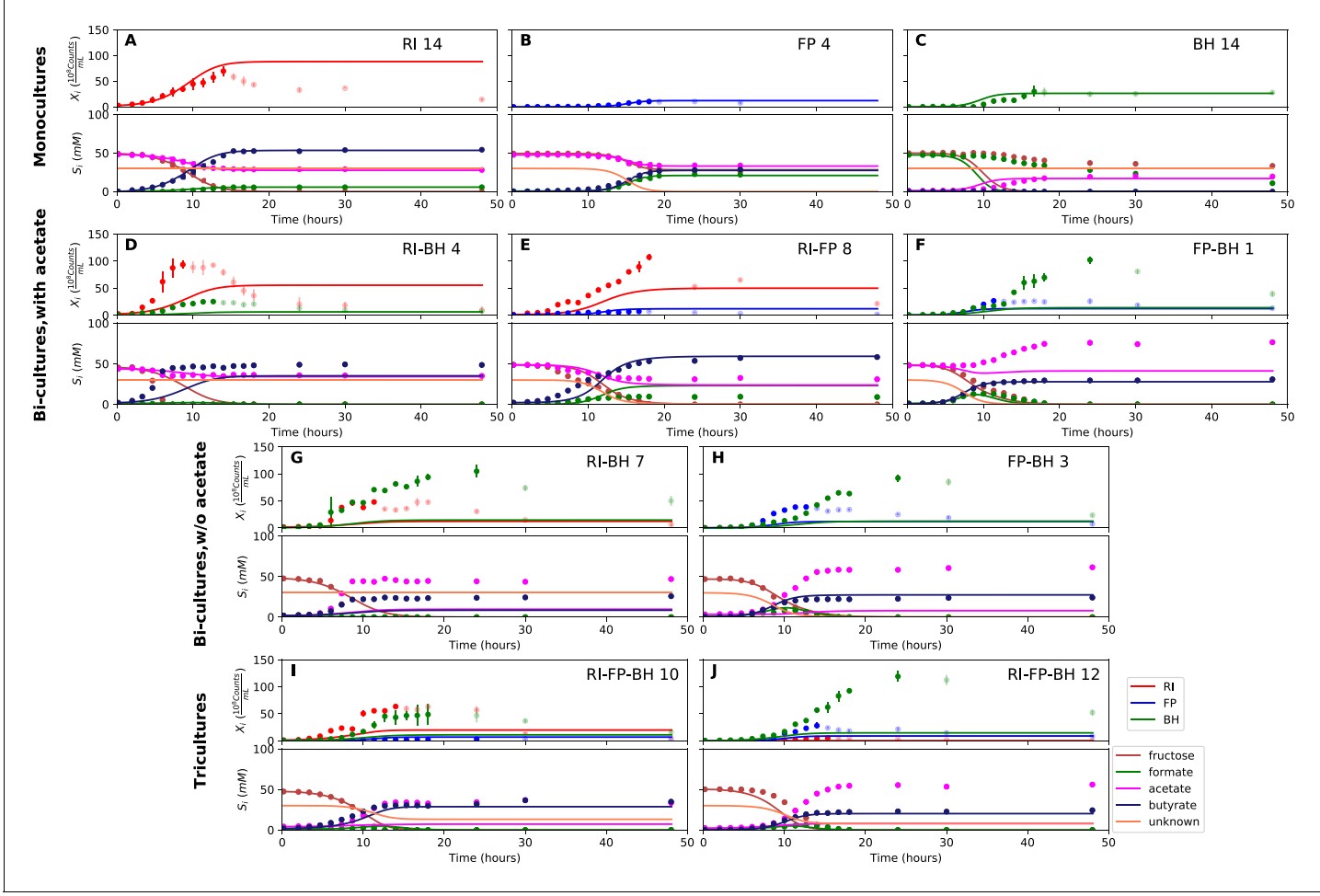

**Figure 4.** Model parameterized on monocultures does not fit co-culture data well. (**A–C**) Fit to monoculture experiments selected for parameterization. (**D–F**) Fit to selected co-culture experiments with initial acetate. (**G–H**) Fit to selected co-culture experiments without initial acetate. (**I–J**) Fit to tri-cultures dominated by *Roseburia intestinalis* L1-82 (RI) and *Blautia hydrogenotrophica* S5a33 (BH) versus *Faecalibacterium prausnitzii* A2-165 (FP) and BH, respectively. Lines represent model predictions and dots represent observations. The whiskers represent technical variation across triplicates. Transparent points indicate declining cell numbers; corresponding samples were not taken into account for model fitting. The unknown compound represents an unspecified co-substrate assumed to be required by FP. Metabolites not included in the model are omitted from the plot. Experiment identifiers indicate which of the biological replicates is displayed. The model was parameterized on experiments RI_8, RI_14, FP_4, FP_15 and BH_14.

DOI: https://doi.org/10.7554/eLife.37090.010

The following source data and figure supplements are available for figure 4:

**Source data 1.** The results of the simulations with the kinetic model using parameterization 1 are provided for each of the fermentation experiments shown in *Figure 4*.

DOI: https://doi.org/10.7554/eLife.37090.014

**Figure supplement 1.** Fit to monoculture experiments for the model parameterized on monocultures only.

DOI: https://doi.org/10.7554/eLife.37090.011

**Figure supplement 2.** Fit to bi-culture experiments for the model parameterized on monocultures only.

DOI: https://doi.org/10.7554/eLife.37090.012

**Figure supplement 3.** Fit to tri-culture experiments for the model parameterized on monocultures only.

DOI: https://doi.org/10.7554/eLife.37090.013

equations implementing a combination of additive and multiplicative Monod functions (see 'Materials and methods'). The model differentiates between substrates required for growth and co-substrates such as acetate that enhanced growth but were not required. It also took strain-specific differences in lag phases into account. As we observed that *F. prausnitzii* A2-165 did not deplete fructose, presumably because of a lack of co-factors, we introduced a dependency on an undefined metabolite referred to as 'unknown compound'.

We parameterized this model on selected monoculture experiments and then predicted monoculture dynamics (*Figure 4A–C*, *Figure 4—figure supplement 1*). The model reached high prediction accuracy for *F. prausnitzii* A2-165 and *R. intestinalis* L1-82, but did not describe well the experimental data for *B. hydrogenotrophica* S5a33 (see *Table 1*). More precisely, the model showed that *B. hydrogenotrophica* S5a33 did not consume formate and fructose as quickly as would be expected if its growth follows Monod kinetics. We confirmed culture homogeneity by analyzing the16S rRNA gene sequencing data of the last sample (*Figure 3—figure supplement 1*). A yeast contaminant (*S. cerevisiae* S288c) that was detected in the RNA-seq data for the *B. hydrogenotrophica* S5a33 monoculture samples (*Figure 3—figure supplement 2*) does not explain the incongruence between growth and energy source consumption, since (i) no contamination was observed on plates inoculated with bioreactor samples and incubated under anaerobic and aerobic conditions, (ii) *S. cerevisiae* would consume fructose, and (iii) no ethanol production was measured. We also found only small concentrations of potential peptide degradation products (isobutyric acid and isovaleric acid). We therefore assumed that *B. hydrogenotrophica* S5a33 in monoculture initially grew on undefined medium components and only later switched to formate and fructose, but the time resolution was insufficient to take this potentially biphasic growth into account.

We also compared model performance for *R. intestinalis* L1-82 with and without product inhibition by hydrogen gas. As we found no differences in model performance, we removed an initial hydrogen gas inhibition term.

## Formate is cross-fed from butyrate producers to *B. hydrogenotrophica* S5a33

When growing *F. prausnitzii* A2-165 and *B. hydrogenotrophica* S5a33 together, we observed that fructose was entirely depleted and that acetate, butyrate, hydrogen gas, carbon dioxide and small concentrations of lactate were produced (*Figure 3F*). Interestingly, there was an initial production of formate, which was then consumed, confirming that formate was cross-fed from *F. prausnitzii* A2-165 to *B. hydrogenotrophica* S5a33. Formate consumption was also observed without initial acetate (*Figure 3H*).

In the bi-culture of *R. intestinalis* L1-82 and *B. hydrogenotrophica* S5a33, carbon dioxide, hydrogen gas, butyrate and small concentrations of lactate were produced, whereas fructose and a small amount of acetate were consumed (*Figure 3D*). The same fermentation products were also obtained in the absence of initial acetate (*Figure 3G*). In contrast to *R. intestinalis* L1-82 in monoculture, no formate was detected in this bi-culture, suggesting that it was entirely cross-fed to *B. hydrogenotrophica* S5a33. It was unclear whether the carbon dioxide and hydrogen gas produced by *R. intestinalis* L1-82 reached concentrations that were sufficient to be cross-fed to *B. hydrogenotrophica* S5a33.

Finally, when *R. intestinalis* L1-82 and *F. prausnitzii* A2-165 were co-cultivated, fructose and acetate were consumed and butyrate, formate, hydrogen gas and carbon dioxide were produced (*Figure 3E*). The finding that formate reached lower concentrations in this co-culture than in *F. prausnitzii* A2-165 monoculture already hints at a negative effect of *R. intestinalis* L1-82 on the growth of *F. prausnitzii* A2-165.

## Comparison of mono- and co-culture data suggests ecological interactions

Since Gause's early work on competition between yeast and *Paramecium* species (*Gause, 1932*; *Gause, 1934*), growth rates in mono- and bi-culture experiments have been compared to determine ecological interactions (e.g. *de Vos et al., 2017*; *Freilich et al., 2011*; *Wang et al., 2017*). The rationale is that the growth rates of mutualistic organisms grown in bi-

culture should increase compared to their growth rates in monoculture, whereas the bi-culture growth rates of competitors should decrease compared to their growth rates in monoculture.

When comparing maximal abundances, cross-feeding and competitive interactions were already apparent. Both *F. prausnitzii* A2-165 and *B. hydrogenotrophica* S5a33 reached significantly higher maximal bacterial counts in *F. prausnitzii* A2-165/*B. hydrogenotrophica* S5a33 bi-cultures and in tri-cultures with *F. prausnitzii* A2-165 dominance (*Figure 3F,H and J*) than they did in monoculture (*Figure 3B and C*), suggesting a mutualistic relationship (unpaired two-sided Wilcoxon *F. prausnitzii* A2-165: shift 0.4, 95% confidence interval 0.12–0.55, p-value 0.03; *B. hydrogenotrophica* S5a33: shift 0.5, 95% confidence interval 0.33–0.69, p-value 0.017). The maximal cell number of *F. prausnitzii* A2-165 tended to be lower when competing with *R. intestinalis* L1-82 (*Figure 3E*) than when grown alone (unpaired two-sided Wilcoxon: shift 0.47, 95% confidence interval −0.03 and 1.42, p-value 0.11). Interestingly, there was no difference in maximal bacterial counts for *R. intestinalis* L1-82 alone versus *R. intestinalis* L1-82 grown with *F. prausnitzii* A2-165 in bi-cultures or in tri-cultures with *R. intestinalis* L1-82 dominance (unpaired two-sided Wilcoxon: shift 0.07, 95% confidence interval −0.39 and 0.31, p-value 0.69), so that formally, their relationship could be described as amensalism (one organism is affected negatively whereas the other is not affected). Finally, according to the maximal bacterial counts, *B. hydrogenotrophica* S5a33 benefited more from the presence of *F. prausnitzii* A2-165 than from that of *R. intestinalis* L1-82 (unpaired two-sided Wilcoxon: shift 0.29, 95% confidence interval 0.06 and 0.93, p-value 0.008).

## Model needs bi-culture data to predict tri-culture dynamics accurately

When growing all three gut bacterial strains together, fructose was consumed and butyrate, acetate, carbon dioxide, hydrogen gas and lactate were produced. Formate was produced initially, peaked between 10 and 15 hr and was below the detection limit after 18 hr of fermentation (*Figure 3I and J*). We performed the tri-culture six times with varying species proportions in the inoculum and found that in all tri-cultures, *B. hydrogenotrophica* S5a33 was always dominant, together with either *R. intestinalis* L1-82 or *F. prausnitzii* A2-165 as co-dominant partner. In two out of the six cases, *R. intestinalis* L1-82 was co-dominant, whereas *F. prausnitzii* A2-165 was co-dominant in the remaining four. The result mattered for the final butyrate concentrations, which averaged 37.5 mM when *R. intestinalis* L1-82 won and 23.5 mM when *F. prausnitzii* A2-165 won.

We attempted to describe tri-culture dynamics with the model parameterized on monocultures, but failed to obtain a good fit (see *Table 1* and *Figure 4—figure supplements 2* and *3*). After a series of tests, we concluded that incorporating bi-culture data was necessary to describe tri-culture dynamics. We finally selected two *F. prausnitzii* A2-165 monocultures and the *R. intestinalis* L1-82/*B. hydrogenotrophica* S5a33 and *F. prausnitzii* A2-165/*B. hydrogenotrophica* S5a33 bi-cultures with initial acetate to parameterize our model. As a validation, we predicted the behavior of *R. intestinalis* L1-82/*B. hydrogenotrophica* S5a33 and *F. prausnitzii* A2-165/*B. hydrogenotrophica* S5a33 bi-cultures without initial acetate, which resulted in a good fit (*Figure 5G and H*, *Figure 5—figure supplement 2*). The model parameterized on mono- and bi-cultures fitted the tri-culture data better than the model parameterized on monocultures only (*Table 1*, *Figure 4I and J*, *Figure 5I and J*, *Figure 4—figure supplement 3* and *Figure 5—figure supplement 3*).

When inspecting the differences between the two parameterizations, we found that the model parameterized on monocultures predicted lower abundances for all three species in bi- and tri-cultures than they actually reached (*Figure 4D–J*, *Figure 4—figure supplements 2* and *3*). Vice versa, the model parameterized on mono- and bi-cultures predicted too high abundances for *R. intestinalis* L1-82 and *B. hydrogenotrophica* S5a33 in monoculture (*Figure 5A and C*, *Figure 5—figure supplement 1*; the *F. prausnitzii* A2-165 monoculture was included in the parameterization). According to the difference in maximal tri-culture cell counts predicted with the two parameterizations, *B. hydrogenotrophica* S5a33 did significantly better in tri-culture than expected on the basis of its monoculture growth (unpaired two-sided Wilcoxon: shift 83, 95% confidence interval 30–92, p-value: 0.002).

The fact that a single model parameterization could not describe well both mono- and tri-culture dynamics is a sign of emergent behavior in the presence of interaction partners. When looking at the parameters inferred from mono- and bi-cultures (given in *Supplementary file 2*), *B. hydrogenotrophica* S5a33's consumption rates for formate and fructose and *R. intestinalis* L1-82's consumption rate for fructose were lower than their values obtained from mono-culture parameterization, whereas their maximal growth rates were not much affected (*B. hydrogenotrophica* S5a33) or increased (*R.*

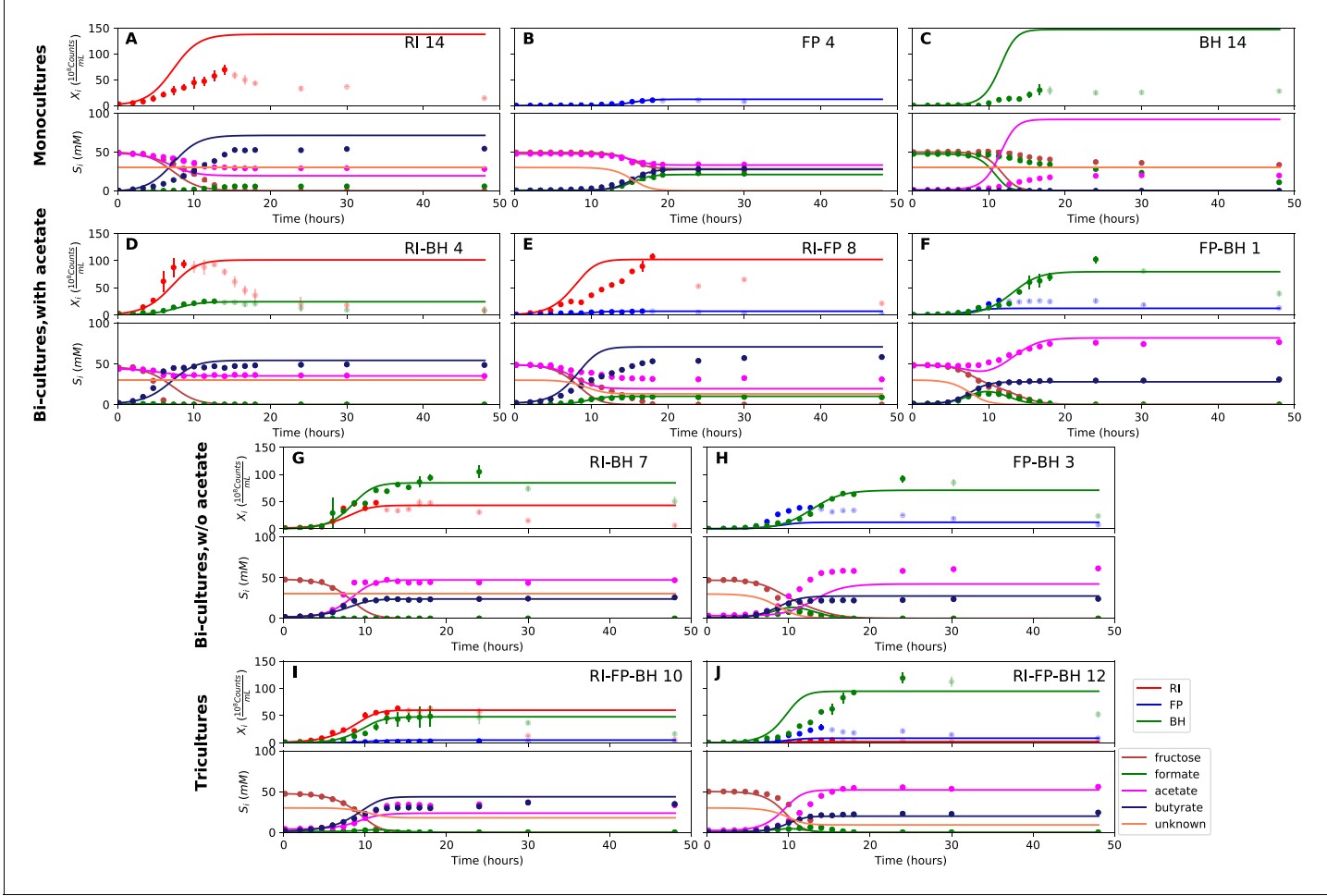

**Figure 5.** Model parameterized on mono- and bi-cultures improves fit to co-culture data as compared to parameterization on monocultures alone. (**A**–**C**) Fit to selected monoculture experiments. (**D**–**F**) Fit to selected co-culture experiments with initial acetate (D and F were included in parameterization). (**G**–**H**) Fit to selected co-culture experiments without initial acetate, which were not part of the parameterization. (**I**–**J**) Fit to tri-cultures dominated by *Roseburia intestinalis* L1-82 (RI) and *Blautia hydrogenotrophica* S5a33 (BH) versus *Faecalibacterium prausnitzii* A2-165 (FP) and BH, respectively. Lines represent model predictions and dots represent observations. The whiskers represent technical variation across triplicates. Transparent points indicate declining cell numbers; corresponding samples were not taken into account for model fitting. The unknown compound represents an unspecified co-substrate assumed to be required by FP. Metabolites not included in the model are omitted from the plot. Experiment identifiers indicate which of the biological replicates is displayed. The model was parameterized on experiments FP_4, FP_15, FP_BH_1, FP_BH_2 and RI_BH_4.

DOI: https://doi.org/10.7554/eLife.37090.015

The following source data and figure supplements are available for figure 5:

**Source data 1.** The results of the simulations with the kinetic model using parameterization 2 are provided for each of the fermentation experiments shown in *Figure 5*.
DOI: https://doi.org/10.7554/eLife.37090.019
**Figure supplement 1.** Fit to monoculture experiments for the model parameterized on selected monocultures and bi-cultures.
DOI: https://doi.org/10.7554/eLife.37090.016
**Figure supplement 2.** Fit to bi-culture experiments for the model parameterized on selected monocultures and bi-cultures.
DOI: https://doi.org/10.7554/eLife.37090.017
**Figure supplement 3.** Fit to tri-culture experiments for the model parameterized on selected monocultures and bi-cultures.
DOI: https://doi.org/10.7554/eLife.37090.018

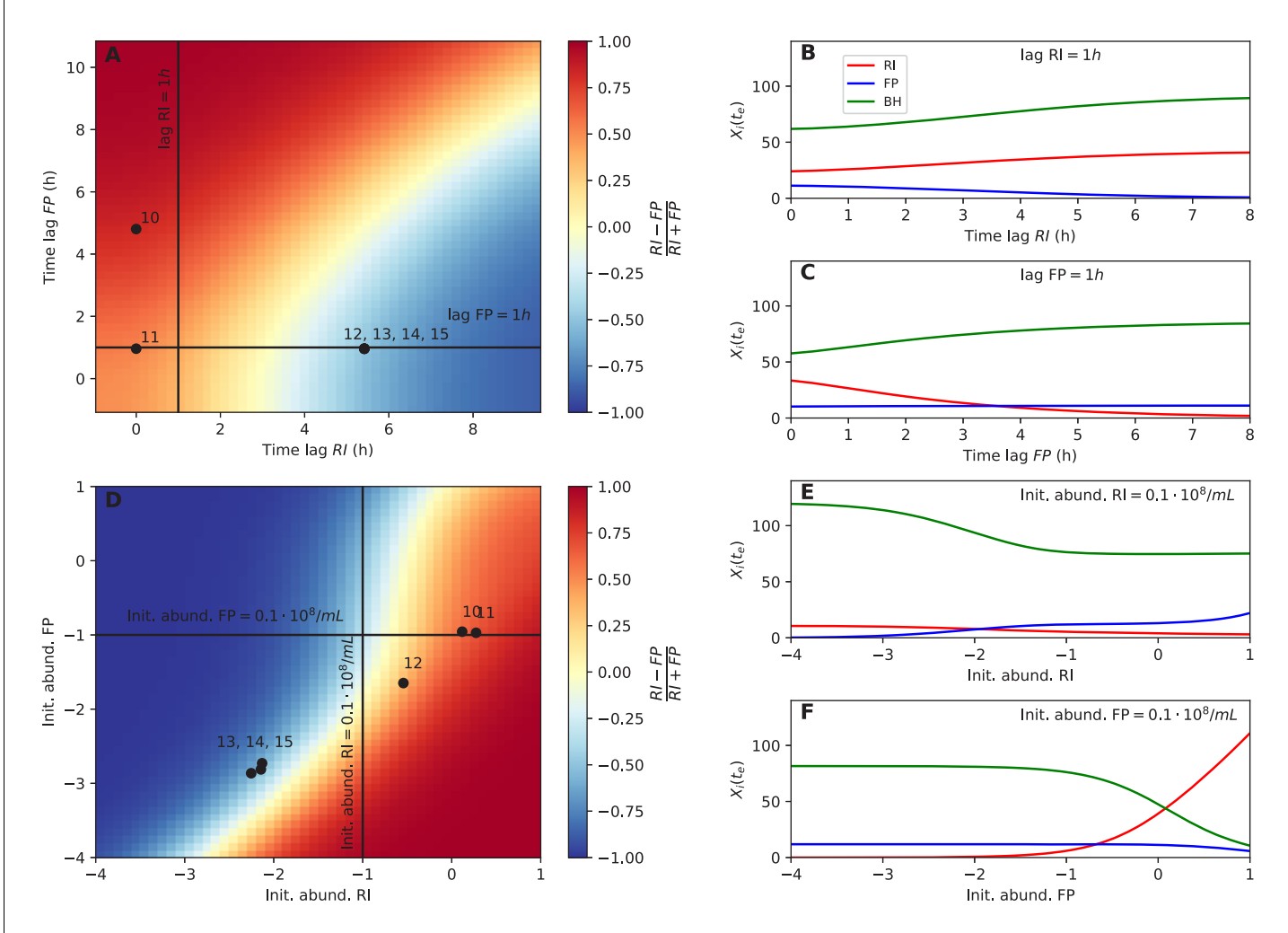

**Figure 6.** Initial abundance and lag phase determine the order of abundance in the final time point of the tri-culture. (A) The tri-culture dynamics is simulated with different lag-phase values for *Faecalibacterium prausnitzii* A2-165 (FP) and *Roseburia intestinalis* L1-82 (RI) and the resulting end point abundance ratio of FP and RI is plotted in a heat map that is colored in blue for FP dominance and in red for RI dominance. The observed tri-culture data (black circles) are plotted according to the estimated experimental lag phases for RI and FP. The predicted RI or FP dominance agrees with the observed dominance in all six cases. (B, C) Simulations illustrate the dependency of the end point abundances ($X_i(t_e)$) of the three strains on the lag phase of RI and FP. (D) The tri-culture dynamics is simulated for varying initial abundances (init. abund.) of FP and RI and their resulting end point abundance ratio is visualized in a heat map. Three of the four FP-dominated experiments (13–15) and both RI-dominated experiments (10 and 11) are situated within their predicted region of dominance. (E, F) The end point abundance $X_i(t_e)$ of the three strains is non-linearly dependent on the initial abundance of RI and FP in simulations, illustrating that dominance in batch is sensitive to initial conditions. All simulations were carried out with the model parameterized on mono- and bi-culture data (parameterization 2). Initial abundances are plotted in logarithmic scale. For the simulations in (A–C), the initial abundances of RI, FP and *Blautia hydrogenotrophica* S5a33 (BH) were set to 0.58, 0.04 and 0.21 $10^8$ counts/mL, respectively, whereas for the simulations in (D–F), the lag phase for RI, FP and BH were set to 0.33, 0.08 and 0.1 h, respectively. These initial abundance and lag-phase values represent the averages of observed initial abundances and estimated lag phases across all tri-culture experiments.

DOI: https://doi.org/10.7554/eLife.37090.020

The following source data and figure supplement are available for figure 6:

**Source data 1.** Predictions for the final abundances of the three strains are provided for different lag-phase values and initial abundances using the kinetic model with parameterization 2.
DOI: https://doi.org/10.7554/eLife.37090.022
**Figure supplement 1.** Sensitivity analysis.
DOI: https://doi.org/10.7554/eLife.37090.021

*intestinalis* L1-82). Thus, according to this analysis, less of the energy source is needed in the presence of an interaction partner than in monoculture.

## Initial abundance and lag phase predict strain dominance in tri-culture

Next, we tested whether dominance in tri-culture could be predicted from lag phase and initial abundance. Towards this aim, we computed the *F. prausnitzii* A2-165/*R. intestinalis* L1-82 ratio in simulations with varying lag phase and initial abundance. Experimental observations of dominance agreed well with the model predictions (*Figure 6A and D*). Our systematic investigation also showed that there was a non-linear relationship between initial *F. prausnitzii* A2-165 abundance and *R. intestinalis* L1-82 dominance (*Figure 6F*). Thus, even when initial abundances, lag phases and species interactions were known, it is hard to predict the winner (and hence the resulting butyrate concentration) intuitively without a model in hand. The final abundances of the three strains in simulations were also non-linearly dependent on other parameters, including *B. hydrogenotrophica* S5a33's growth rate, its fructose consumption rate and its fructose half-saturation constant (*Figure 6—figure supplement 1*). These results underline that, in addition to kinetic parameters, initial conditions and lag phase can determine strain abundances in co-culture in a non-linear way.

## Altered gene expression in response to interaction partners provides first insights into emergent behavior

To further investigate the emergent behavior, we sequenced RNA for three time points and two biological replicates for each of the three monocultures and for the tri-culture with *F. prausnitzii* A2-165 co-dominance, and assessed significantly differential gene expression across all samples in mono- versus tri-cultures for all three strains (*Supplementary file 3*). In total, 9.3%, 10.9% and 7.0% of *R. intestinalis* L1-82's, *F. prausnitzii* A2-165's and *B. hydrogenotrophica* S5a33's protein-coding genes were significantly differentially expressed (protein numbers taken from UniProt; *The UniProt Consortium, 2017*).

Interestingly, in tri-culture, *F. prausnitzii* A2-165 downregulated a series of enzymes needed for vitamin $B_{12}$ coenzyme biosynthesis. Cobalamin (vitamin $B_{12}$) was one of the co-factors suspected to limit *F. prausnitzii* A2-165 growth in monoculture, and this finding could mean that *F. prausnitzii* A2-165 benefited from greater co-factor availability in tri-culture. We tested whether *F. prausnitzii* A2-165 grown in test tubes benefited from added vitamin $B_{12}$ (*Supplementary file 4*) but did not see a significant increase in cell numbers. Although this indicates that *F. prausnitzii* A2-165 downregulates the $B_{12}$ production pathway upon presumably higher cobalamin availability in the tri-culture, this does not explain the change in growth characteristics.

In tri-culture, *F. prausnitzii* A2-165 also upregulated enzymes that are involved in amino acid and oligopeptide transport and amino acid and protein biosynthesis. *B. hydrogenotrophica* S5a33 likewise upregulated amino acid biosynthesis in tri-culture. For *R. intestinalis* L1-82, which reached lower abundances in the selected tri-cultures than in monoculture, the transcription response was mixed: some amino acid biosynthesis enzymes were downregulated, others upregulated (including enzymes involved in ornithine biosynthesis). However, the expression of ribosomal proteins was lower than that in *R. intestinalis* L1-82 mono-culture, in agreement with its long lag phase in the selected tri-cultures. In summary, the analysis of differential gene expression uncovered a number of metabolic changes in the presence of interaction partners, thus further supporting the altered behavior detected through modeling.

## Discussion

Here, we investigated the dynamics of a well-defined, small, but representative synthetic gut microbial community, consisting of the three strains *B. hydrogenotrophica* S5a33, *F. prausnitzii* A2-165 and *R. intestinalis* L1-82. We found that *B. hydrogenotrophica* S5a33 is metabolically versatile and grew as fast as primary fermenters such as *R. intestinalis* L1-82. We demonstrated experimentally that formate was cross-fed between *B. hydrogenotrophica* S5a33 on the one hand and *F. prausnitzii* A2-165 and *R. intestinalis* L1-82 on the other, and confirmed mutualistic as well as competitive interactions between these three bacterial strains. When growing on formate, we identified *B. hydrogenotrophica* S5a33 as a net producer of both hydrogen gas and carbon dioxide, in contrast with its traditionally assumed role in the gut ecosystem. Although formate is rarely highlighted as a key

intermediate in gut cross-feeding interactions, it has been reported to be an end-product of primary polysaccharide degradation by both *Bifidobacterium* and *Lactobacillus* species (*Falony et al., 2009b*; *Moens et al., 2017*). Hence, our results invite a re-evaluation of the ecological niche of *B. hydrogenotrophica* in relation to its potential for microbial formate production.

Although only one strain was tested for each of the three colonic species considered, the relationships described here probably generalize to species level. Each of the strains in the species descriptions (including four additional *R. intestinalis* strains and one additional strain each of *F. prausnitzii* and *B. hydrogenotrophica*) are reported to produce and consume the same key metabolites as the strain selected for our experiments (*Bernalier et al., 1996*; *Duncan et al., 2002a*; *Duncan et al., 2002b*). In addition, three further *F. prausnitzii* strains (SL3/3, KLE1255 and M21/2) have been predicted to produce formate (*Magnúsdóttir et al., 2017*) and two additional *R. intestinalis* strains (M50/1 and XB6B4) contain pyruvate ferredoxin oxidoreductase, an enzyme assumed to be crucial for hydrogen gas production in Clostridial cluster XIVa (*Falony et al., 2009c*). In addition, *Rey et al. (2010)* demonstrated that in gnotobiotic mice, *B. hydrogenotrophica* S5a36 grows better in the presence of a primary fermenter (*Bacteroides thetaiotaomicron*) than alone. However, further experiments are required to test whether the potential for cross-feeding is realized in other strain combinations.

The model, which encodes our knowledge of the system, is important not only for predictions but also as a reference. We gained insights from its agreements as well as from its disagreements with our observations. For instance, we assumed initially that *R. intestinalis* L1-82 would be inhibited by the hydrogen gas it generated. However, a hydrogen gas inhibition term did not improve the accuracy of predicted *R. intestinalis* L1-82 behavior in monoculture, which suggested that hydrogen gas inhibition did not affect *R. intestinalis* L1-82 growth at the concentrations reached in our experiments. We also needed the model to ascertain that changes from mono- to bi- or tri-cultures were not just the result of variations in the inoculum composition or the lag phase, but that there was a true change in the dynamics that the model parameterized on monocultures alone could not capture. We confirmed this emergent behavior with RNA-seq, which revealed significantly different gene expression in tri-culture than in mono-culture, especially for *F. prausnitzii* A2-165 and *B. hydrogenotrophica* S5a33. The downregulation of *F. prausnitzii* A2-165's vitamin $B_{12}$ coenzyme biosynthesis pathway in tri-culture is of particular interest, as it suggests that dependency on co-factors changes with interaction partners. It has been posited that the majority of gut microorganisms in need of $B_{12}$ precursors are unable to synthesize them (*Degnan et al., 2014*). If this need is altered by the presence of interaction partners, it cannot be exploited as easily for selective manipulation as suggested by *Degnan et al. (2014)*.

Although kinetic models parameterized on monocultures may in some cases describe bi-culture dynamics correctly (*Van Wey et al., 2014*), our example shows that this is not a general property. This means that models of microbial communities will have to take the internal metabolism of community members and their response to interaction partners into account. Gut bacteria such as *B. hydrogenotrophica* S5a33 have flexible metabolic strategies that they employ according to circumstances. Emergent metabolism in the presence of interaction partners has been described in theoretical work before (*Chiu et al., 2014*), but has been rarely investigated experimentally (for example, in *Aharonovich and Sher, 2016*). Constraint-based modeling approaches, which can take emergent metabolism into account (*Orth et al., 2010*), require high-quality metabolic reconstructions for each community member, which take months of curation effort to obtain (*Thiele and Palsson, 2010*). Thus, scaling strain-level quantitative models to larger communities will be a formidable challenge.

Mono- and bi-cultures are increasingly carried out in batch in a high-throughput fashion to determine ecological interactions and to quantify their strengths (*de Vos et al., 2017*; *Sher et al., 2011*). Such systematic quantification is an important step forward, but there are challenges to tackle. Our work showed that dominance in batch may sensitively (i.e., non-linearly) depend on initial conditions such as the lag phase and the initial abundance, both of which are hard to control experimentally. Thus, a growth experiment performed in biological replicates but with the same inoculum may identify one strain as the winner and another as the loser. However, a replicate with a slightly different inoculum composition may provide results that support the opposite conclusion. Such a dependency on the initial conditions (albeit with larger abundance differences) has also been reported in several competition experiments for *Streptomyces* species (*Wright and Vetsigian, 2016*) and may thus be a common case.

To ascertain that bacteria change their behavior in response to an interaction partner, RNA-seq is carried out on mono- and bi-cultures (*Aharonovich and Sher, 2016*; *González-Torres et al., 2015*; *Plichta et al., 2016*). Here, we showed that a model can also reveal emergent behavior by its failure to describe co-culture dynamics when parameterized on monocultures only. How such approaches could be scaled up to achieve the high-throughput needed for systematic measurements of interaction strengths is an open question.

Although the combination of mutualism and competition has been explored in vitro previously (*Rivière et al., 2015*), to the best of our knowledge, this is the first investigation of a defined bacterial community in which two strain pairs mutually cross-feed and compete. In the case of *F. prausnitzii* A2-165 and *B. hydrogenotrophica* S5a33, mutualism appears to supersede competition, leading to increased maximal bacterial numbers coupled with upregulation of biosynthesis for both interaction partners. For *R. intestinalis* L1-82 and *B. hydrogenotrophica* S5a33, we had no such clear experimental evidence, as RNA-seq was performed on tri-cultures dominated by *F. prausnitzii* A2-165, but the comparison of maximum bacterial numbers across mono-, bi- and tri-cultures suggested that *R. intestinalis* L1-82 and *B. hydrogenotrophica* S5a33 did not benefit as much from each other as *F. prausnitzii* A2-165 and *B. hydrogenotrophica* S5a33 did. As the model described *R. intestinalis* L1-82/*B. hydrogenotrophica* S5a33 bi-culture dynamics well without taking carbon dioxide and hydrogen gas cross-feeding into account, we assume that because of their low partial pressure, gasses were less efficiently cross-fed to *B. hydrogenotrophica* S5a33 than formate, although both are probably metabolized via the same pathway (Wood-Ljungdahl). Thus, interactions that look similar on paper can play out differently, depending on the environment.

In the two replicates of the *R. intestinalis* L1-82/*F. prausnitzii* A2-165 bi-culture, *R. intestinalis* L1-82 and *F. prausnitzii* A2-165 both survived (albeit *F. prausnitzii* A2-165 in far lower numbers) despite competing for the same substrate. This is presumably because in our experimental set-up, the time until nutrient depletion was too short for the competitive exclusion principle to apply. Our tri-culture experiments also demonstrated the importance of initial conditions in determining fermentation end-products. According to our model, the initial abundance and lag phase determined whether butyrate reached high or low concentrations in the tri-culture fermentations. Since these are likely to be relevant parameters in the gut environment and difficult to control, cocktail communities will have to be designed such that they will carry out their function across a wide range of initial conditions.

Although our work highlighted a number of challenges to microbial community modeling, the model's ability to predict tri-culture dynamics from bi-cultures gives hope that, with sufficient knowledge, we will ultimately be able to model more complex microbial communities.

## Materials and methods

### Key resources table

| Reagent type (species) or resource | Designation | Source or reference | Identifiers | Additional information |
|---|---|---|---|---|
| Strain (*Roseburia intestinalis*), strain background (human feces) | *Roseburia intestinalis* L1-82 | Deutsche Sammlung von Mikroorganismen und Zellkulturen (DSMZ) | DSM_14610 | |
| Strain (*Faecalibacterium prausnitzii*), strain background (human feces) | *Faecalibacterium prausnitzii* A2-165 | Deutsche Sammlung von Mikroorganismen und Zellkulturen (DSMZ) | DSM_17677 | |
| Strain (*Blautia hydrogenotrophica*), strain background (human feces) | *Blautia hydrogenotrophica* S5a33 | Deutsche Sammlung von Mikroorganismen und Zellkulturen (DSMZ) | DSM_10507 | |
| Sequence-based reagent | TaqMan primer and probe for *Roseburia intestinalis* L1-82 | This paper | | Sequences provided in *Supplementary file 5* |

*Continued on next page*

*Continued*

| Reagent type (species) or resource | Designation | Source or reference | Identifiers | Additional information |
|---|---|---|---|---|
| Sequence-based reagent | TaqMan primer and probe for *Faecalibacterium prausnitzii* A2-165 | This paper | | Sequences provided in *Supplementary file 5* |
| Sequence-based reagent | TaqMan primer and probe for *Blautia hydrogenotrophica* S5a33 | This paper | | Sequences provided in *Supplementary file 5* |
| Commercial assay or kit | Phenol-Free Total RNA Purification Kit | VWR International | N788-KIT | Coupled with DNase I treatment |
| Commercial assay or kit | TURBO DNA-free KitTURBO DNase Treatment and Removal Reagents | Ambion (ThermoFisher Scientific) | AM1907 | |
| Commercial assay or kit | RNA Clean and Concentrator (RCC)–25 Kit | Zymo Research | R1017 | |
| Commercial assay or kit | Qubit dsDNA HS Assay Kit | ThermoFisher Scientific | Q32854 | |
| Commercial assay or kit | Agilent RNA 6000 Pico Kit | Agilent Technologies | 5067–1513 | |
| Commercial assay or kit | Agilent RNA 6000 Nano Kit | Agilent Technologies | 5067–1511 | |
| Commercial assay or kit | Ribozero rRNA removal for Gram-positive bacteria | Illumina | MRZGP126 | |
| Commercial assay or kit | Illumina TruSeq stranded mRNA library preparation kit | Illumina | RS-122–2101 | Library preparation was performed without the mRNA purification step |

## Microorganisms and media

Human isolates of *Roseburia intestinalis* A2-165 (DSM 14610[T]), *Faecalibacterium prausnitzii* L1-82 (DSM 17677[T]) and *Blautia hydrogenotrophica* S5a33 (DSM 10507[T]) were obtained from the Deutsche Sammlung von Mikroorganismen und Zellkulturen (DSMZ, Germany) and stored at −80˚C in reinforced clostridial medium (RCM; Oxoid Ltd., Basingstoke, United Kingdom), supplemented with 25% (vol/vol) glycerol as a cryoprotectant.

A recently published medium for colon bacteria (mMCB) that allows growth of *F. prausnitzii* A2-165 (*Moens et al., 2016*) was modified by adding nitrogen sources and trace elements as detailed below. This medium had the following composition (concentrations in g $L^{-1}$): bacteriological peptone (Oxoid), 6.5; soy peptone (Oxoid), 5.0; yeast extract (VWR International, Darmstadt, Germany), 3.0; tryptone (Oxoid), 2.5; NaCl (VWR International), 1.5; $K_2HPO_4$ (Merck, Darmstadt, Germany), 1.0; $KH_2PO_4$ (Merck), 1.0; $Na_2SO_4$ (VWR International), 2.0; $MgSO_4 \cdot 7H_2O$ (Merck), 1.0; $CaCl_2 \cdot 2H_2O$ (Merck), 0.1; $NH_4Cl$ (Merck), 1.0; cysteine-HCl (Merck), 0.4; $NaHCO_3$ (VWR International), 0.2; $MnSO_4 \cdot H_2O$ (VWR International), 0.05; $FeSO_4 \cdot 7H_2O$ (Merck), 0.005; $ZnSO_4 \cdot 7H_2O$ (VWR International), 0.005; hemin (Sigma-Aldrich, Steinheim, Germany), 0.005; menadione (Sigma-Aldrich), 0.005; and resazurin (Sigma-Aldrich), 0.001. The medium was supplemented with 1 mL $L^{-1}$ of selenite and tungstate solution (NaOH (Merck), 0.5; $Na_2SeO_3 \cdot 5H_2O$ (Merck), 0.003; $Na_2WO_4 \cdot 2H_2O$ (Merck), 0.004 and 1 L of distilled water) and 1 mL $L^{-1}$ of trace element solution SL-10 (HCl (Merck, 25%, vol/vol; 7.7 M), $FeCl_2 \cdot 4H_2O$ (Merck), 1.5; $ZnSO_4 \cdot 7H_2O$ (VWR International), 0.148; $MnSO_4 \cdot H_2O$ (VWR International), 0.085; $H_3BO_3$ (Merck), 0.006; $CoCl_2 \cdot 6H_2O$ (Merck), 0.19; $CuSO_4 \cdot 5H_2O$ (VWR International), 0.0034; $NiCl_2 \cdot 6H_2O$ (Merck), 0.024; and $Na_2MoO_4 \cdot 2H_2O$ (Merck), 0.036. Acetate (50 mM or 6.8 g $L^{-1}$ of $CH_3COO^-Na^+3H_2O$ (Merck)) was added to the medium for the mono-culture fermentations with *R. intestinalis* L1-82 and *F. prausnitzii* A2-165, whereas formate (50 mM or 3.4 g $L^{-1}$ of $HCOO^-Na^+$ (VWR International)) was added to the medium for the mono-culture fermentations with *B. hydrogenotrophica* S5a33. The pH of the medium was adjusted to 6.8 and the medium was

autoclaved at 210 kPa and 121°C for 20 min. After sterilization, D-fructose (Merck) was added as the sole energy source aseptically, at a final concentration of 50 mM fructose using sterile stock solutions obtained through membrane filtration using Minisart filters (pore size, 0.2 μm (Sartorius, Göttingen, Germany)).

## Cultivation experiments in stationary bottles

Mono-culture cultivation experiments for *B. hydrogenotrophica* S5a33 were performed in stationary glass bottles without controlling the pH (screening). The bottles contained 50 mL of heat-sterilized pH 6.8 mMCB medium, supplemented with 50 mM of D-fructose (Merck), D-glucose (Merck), D-galactose (Merck), L-fucose (Merck), sodium formate (VWR International), sodium acetate trihydrate (Merck), DL lactic acid (VWR International), oligofructose (Raftilose P95; Beneo-Orafti NV, Tienen, Belgium) or inulin (OraftiHP; Beneo-Orafti) as energy sources (*Falony et al., 2009b*). Additional cultivation experiments were performed in medium devoid of any main energy source to test autotrophic growth. For the cultivation experiments in bottles, stock solutions of fructose, glucose, galactose, fucose, sodium formate, sodium acetate trihydrate, and lactic acid were initially made anaerobically through autoclaving at 210 kPa and 121°C for 20 min. The solutions were subsequently filter-sterilized and transferred into glass bottles, which were sealed with butyl rubber septa that were pierced with a Sterican needle (VWR International) connected with a Millex-GP filter (Merck) to assure sterile conditions. For the cultivation experiments with lactate, the pH was adjusted to 6.8 under anaerobic conditions, using sterile solutions of sodium hydroxide (Merck). Stock solutions of oligofructose and inulin were made sterile by membrane filtration. The inocula were prepared as follows. Cells of the strains under study were transferred from −80°C to test tubes containing 10 mL of RCM that were incubated anaerobically at 37°C for 24 hr. Subsequently, the strains were propagated for 12 hr in glass bottles containing 50 mL of heat-sterilized pH 6.8 mMCB medium, supplemented with the energy source under study, always at a final concentration of 50 mM of fructose equivalents. These pre-cultures were finally added to the glass bottles aseptically. During the inoculum build-up, the transferred volume was always 5% (vol/vol). All bottles were incubated anaerobically at 37°C in a modular atmosphere-controlled system (MG anaerobic work station; Don Withley Scientific Ltd., West Yorkshire, United Kingdom) that was continuously sparged with a mixture of 80% $N_2$, 10% $CO_2$, and 10% $H_2$ (Air Liquide, Paris, France). Samples for further analyses were withdrawn after 0, 6, 12, 24, 48, and 100 hr. All experiments were performed at least in duplicate.

## Fermentation experiments

To prepare inocula, cells were transferred from −80°C to test tubes containing 10 mL of RCM, and incubated at 37°C for 24 hr. Subsequently, the strains were propagated twice for 12 hr in glass bottles containing 100 mL of mMCB medium (with acetate in the case of *R. intestinalis* L1-82 and *F. prausnitzii* A2-165, and with formate in the case of *B. hydrogenotrophica* S5a33), supplemented with fructose. All incubations were performed anaerobically in a modular atmosphere-controlled system (MG anaerobic workstation) that was continuously sparged with a mixture of 80% $N_2$, 10% $CO_2$, and 10% $H_2$ (Air Liquide). The inocula were finally added aseptically to the fermentors. During the inoculum build-up, the transferred volume was always 5% (vol/vol). Fermentations were carried out in 2 L Biostat B-DCU fermentors (Sartorius) containing 1.5 L of mMCB medium supplemented with the co-substrates (acetate and/or formate) if necessary and 50 mM of D-fructose as the energy source. Anaerobic conditions during fermentations were assured by continuously sparging the medium with $N_2$ (PraxAir, Schoten, Belgium) at a flow rate of 70 mL min$^{-1}$. The fermentation temperature was kept constant at 37°C. A constant pH of 6.8 was imposed and controlled automatically, using 1.5 M solutions of NaOH and $H_3PO_4$. To keep the medium homogeneous, a gentle stirring of 200 rpm was applied. Temperature, pH, and agitation speed were controlled online (MFCS/win 2.1 software, Sartorius). Fermentations were followed for 48 hr, with samples taken at 10 min and 2 hr, 3 hr, 5 hr, 6 hr, 7 hr, 9 hr, 10 hr, 11 hr, 13 hr, 14 hr, 15 hr, 17 hr, 18 hr, 24 hr, 30 hr and 48 hr after inoculation. At selected time points (3 hr, 9 hr and 15 hr after inoculation), subsamples were treated for RNA extraction by adding 5 vol of RNAlater (Thermo Fisher Scientific). All mono- and tri-culture fermentations were performed in triplicate. All bi-culture fermentations were performed in duplicate, except for the bi-culture fermentation using medium lacking acetate with *F. prausnitzii* A2-165 and *B. hydrogenotrophica* S5a33, which was performed only once.

## Addition of vitamin B$_{12}$ to *F. prausnitzii* A2-165

*F. prausnitzii* A2-165 was grown in test tubes containing 10 mL of RCM each. Then 10, 50 and 100 µL of filter-sterilized (0.22 µm, Merck Millipore) 0.1 g/L vitamin B$_{12}$ solution (Sigma-Aldrich) was added to reach a final concentration of 0.1, 0.5 and 1 mg/L, respectively, in the test tubes. For each of the three concentrations as well as for the control (without added B$_{12}$), bacterial abundance was followed in three test tubes. Samples were taken after 24 hr and 48 hr and cell counts were obtained via flow cytometry as described below.

## Quantification of bacterial abundance

During all experiments, the optical density at 600 nm (OD$_{600}$) was measured against ultrapure water as blank with a VIS spectrophotometer (Genesys 20; Thermo Scientific, Waltham, MA, USA). Each measurement was performed in triplicate. Total bacterial abundance was also measured by flow cytometry, using an Accuri C6 flow cytometer (BD Biosciences, Erembodegem, Belgium), as described previously (*Moens et al., 2016*). All samples were diluted in filter-sterilized water (Vittel, France) to obtain a concentration between $1.0 \times 10^3$ and $5.0 \times 10^6$ cells mL$^{-1}$. Flow cytometric analysis was performed by mixing 500 µL of sample with 5 µL of a 100 $\times$ SYBR Green I solution (Sigma-Aldrich) and 5 µL of a 500 mM ethylenediaminetetra acetic acid (EDTA) solution (Sigma-Aldrich). Afterwards, samples were left in the dark at room temperature for 15 min. Flow cytometric counts were obtained using an Accuri C6 flow cytometer (BD Biosciences), equipped with a 50 mW solid state laser (488 nm). Green fluorescence was measured in the FL1 channel (530 $\pm$ 15 nm) and all data were processed with the Cflow Plus software (Accuri). Gating was performed to distinguish signals from noise. All data were collected as a FL1/SSC density plot with a primary threshold of 10,000 on the FL1 channel. Measurements were performed in triplicate.

qPCR assays with strain-specific TaqMan primers and probes were performed to quantify the abundance of each strain separately. For this, 2 mL of fermentation sample was centrifuged at 20,570 $\times$ *g* for 20 min. Cells were washed in 2 mL of physiological solution (NaCl, 8.5 g L$^{-1}$) and centrifuged again at 20,570 $\times$ *g* for 20 min to obtain washed cell pellets. Subsequently, these cell pellets were resuspended in 2 mL of physiological solution and diluted 20 times for DNA extraction. Direct DNA extractions by alkaline thermal lysis were performed on the basis of the methods used by *Girish et al. (2013)* and *Rudbeck and Dissing (1998)*, modified as follows: 100 µL of the sample was mixed with 100 µL of 0.2 M NaOH in a sterile microcentrifuge tube. The mixture was vortexed and heated at 90℃ for 10 min, after which eight volumes (1600 µL) of 0.04 M Tris HCl pH 7.5 (Thermo Fisher Scientific) was added for pH neutralization. 4 µL of the final mixture was used for qPCR. The extracted genomic DNA was stored at $-20$℃ until qPCR amplification.

Calibration curves were obtained by initially growing all strains in RCM for 24 hr, and two-fold propagation in medium for 12 hr, as described above. From each of these grown cultures, separate four-fold decimal and nine-fold binary dilution series were prepared. The generation of cell pellets, direct extraction of DNA, and subsequent quantification of cell concentrations by flow cytometry were performed as described above, with the exception that, prior to DNA extraction, samples for calibration were diluted less than the fermentation samples, to accommodate a wider qPCR quantification range.

Primers and oligoprobes (listed in *Supplementary file 5*) were manually designed using the online Primer3Plus software (*Untergasser et al., 2007*) and the genome sequences of the strains. Melting temperatures and the presence of hairpins, self-dimers, and pair-dimers were double-checked using the online OligoCalc software (*Kibbe, 2007*). Secondary structures of the generated amplicons were investigated using the online Mfold program (*Zuker, 2003*). Primers and probes were synthesized by Thermo Fisher Scientific. The strain specificities of primers and probes were confirmed in silico by Primer-BLAST (*Ye et al., 2012*) and in vitro by PCR and qPCR analysis on genomic DNA of the strains (*Supplementary file 5*). qPCR assays were carried out using a 7500 FAST Real-Time PCR system (Applied Biosystems, Carlsbad, CA, USA), equipped with 96-well plates. Each qPCR assay mixture of 20 µL contained 10.0 µL of TaqMan Fast Universal PCR Master Mix (2X), no AmpErase UNG (Thermo Fisher Scientific), 2.0 µL of each primer (3.0 µM), 2.0 µL of the TaqMan probe (1.5 µM), and 4.0 µL of extracted genomic DNA solution or sterile nuclease-free water (Thermo Fisher Scientific). The qPCR amplification program consisted of an initial denaturation step at 95℃ for 20 s, followed by 45 two-step cycles at 95℃ for 3 s and at 60℃ for 30 s. In each run,

negative (sterile nuclease-free water without genomic DNA) and positive controls (with extracted genomic DNA from the relevant strains) were used. The cycle threshold (Ct) values were determined using the automatically determined thresholds from the 7500 software v2.0.6 (Applied Biosystems). Finally, during a re-analysis of all qPCR runs, Ct values were normalized using an inter-plate calibrator to account for differences among qPCR runs. The above-described generation of cell pellets, direct extraction of DNA, and qPCR assays were performed in triplicate.

Contamination was checked by aerobic and anaerobic plating on RCM agar and 16S rRNA gene amplicon sequencing of end point fermentation samples (48 hr). Sequencing was performed as described previously (*D'hoe et al., 2018*).

## Metabolite profiling

Concentrations of fructose, as well as concentrations of formate, acetate, butyrate, lactate and ethanol, were determined through high-performance liquid chromatography (HPLC) with refractive index detection, using a Waters chromatograph (Waters, Milford, MA, USA) equipped with an ICSep ICE ORH-801 column (Transgenomic North America, Omaha, NE, USA), and applying external standards, as described previously (*Falony et al., 2009b*). Briefly, the mobile phase consisted of 5 mM $H_2SO_4$ at a flow rate of 0.4 mL $min^{-1}$. The column temperature was kept constant at 35°C. Sample preparation involved a first centrifugation (4618 x *g* for 20 min at 10°C) for removal of cells and debris, followed by the addition of an equal volume of 20% (mass/vol) trichloroacetic acid for protein removal. For determining oligofructose and inulin consumption, samples were incubated at room temperature for 24 hr to assure complete hydrolysis of the polysaccharides. Subsequently, the samples were centrifuged (21,912 x *g*, 20 min, 4°C) and filtered (pore size of 0.2 μm; Uniflo 13 Filter Unit; GE Healthcare, Little Chalfont, UK), prior to injection (30 μL) into the column. Samples were analyzed in triplicate.

Concentrations of hydrogen gas and carbon dioxide in the fermentor gas effluents were determined online through gas chromatography (GC) with thermal conductivity detection (TCD), using a CompactGC (Interscience, Breda, The Netherlands) equipped with a 10 m Molsieve 5A column (hydrogen gas (Varian, Palo Alto, CA, USA)) and a 10 m PoraBOND Q column (carbon dioxide (Varian)), and applying external standards, as described previously (*Falony et al., 2009b*).

For an additional screening experiment with *B. hydrogenotrophica* S5a33 grown in the presence of 350 mM formate, the concentrations of ethanol, acetoin, acetic acid, propionic acid, butyric acid, isobutyric acid and isovaleric acid produced were determined by gas chromatography with flame ionization detection (GC-FID), using a FocusGC chromatograph (Interscience) equipped with a Stabilwax-DA column (Restek, Bellefonte, PA, USA), and applying external standards, as described previously (*Moens et al., 2014*). The samples were analyzed in triplicate.

## Model definition

We modeled change of species abundances over time with the following three ordinary differential equations:

$$\frac{dX_{RI}}{dt} = \Phi_{RI}\left(Q_{RI}, S_{fructose}, S_{acetate}\right)X_{RI}$$

$$\frac{dX_{FP}}{dt} = \Phi_{FP}\left(Q_{FP}, S_{unknown}, S_{fructose}, S_{acetate}\right)X_{FP}$$

$$\frac{dX_{BH}}{dt} = \Phi_{BH}\left(Q_{BH}, S_{fructose}, S_{formate}\right)X_{BH}$$

where $X$ denotes species abundance, $S$ metabolite concentration and $Q$ a lag phase parameter. The growth rates are then defined as non-linear growth functions as described by *Grivet (2001)* and *Smith and Waltman (1995)*, and assuming Monod kinetics (*Monod, 1950*):

$$\Phi_{RI}\left(Q_{RI}, S_{fructose}, S_{acetate}\right) = \Gamma_{RI}(Q_{RI})\mu_{RI} \frac{S_{fructose}}{K_{RI\_fructose} + S_{fructose}} \left(1 + \omega_{RI} \frac{S_{acetate}}{K_{RI\_acetate} + S_{acetate}}\right)$$

$$\Phi_{FP}\left(Q_{FP}, S_{unknown}, S_{fructose}, S_{acetate}\right)$$
$$= \Gamma_{FP}(Q_{FP})\mu_{FP} \frac{S_{unknown}}{K_{FP\_unknown} + S_{unknown}} \frac{S_{fructose}}{K_{FP\_fructose} + S_{fructose}} \left(1 + \omega_{FP} \frac{S_{acetate}}{K_{FP\_acetate} + S_{acetate}}\right)$$

$$\Phi_{BH}\left(Q_{BH}, S_{fructose}, S_{formate}\right) = \Gamma_{BH}(Q_{BH})\mu_{BH}\left(\frac{S_{fructose}}{K_{BH\_fructose} + S_{fructose}} + \omega_{BH}\frac{S_{formate}}{K_{BH\_formate} + S_{formate}}\right)$$

where $K$ is the Monod (half-saturation) constant, $\mu$ is the maximal specific growth rate and $\omega$ a weight parameter. Nutrient dependency can be either obligatory (growth without nutrient is not possible) or facultative (growth without nutrient is possible). For instance, the fructose uptake is multiplied with *R. intestinalis* L1-82's maximal growth rate, whereas its acetate uptake is modeled with an additive term. Therefore, in the absence of fructose, *R. intestinalis* L1-82's growth rate is zero, but this is not the case when acetate is absent. The weight parameter adjusts how strongly a facultative substrate contributes to the overall growth rate. The unknown compound models the dependency of *F. prausnitzii* A2-165 on an undetermined co-factor.

The lag phase function is defined as in *Baranyi and Roberts (1994)*:

$\Gamma_i(Q_i) = \frac{Q_i}{1+Q_i}$,

where $i$ stands for *R. intestinalis* L1-82, *F. prausnitzii* A2-165 or *B. hydrogenotrophica* S5a33.

The $Q_i$ variables follow exponential growth:

$$\frac{dQ_i}{dt} = \mu_i Q_i$$

Thus, the larger the initial value of $Q_i$, the shorter the lag phase.

The changes of metabolite concentrations are then modeled as follows:

$$\frac{dS_{fructose}}{dt} = -\nu_{RI,fructose}\Phi_{RI}X_{RI} - \nu_{FP,fructose}\Phi_{FP}X_{FP} - \nu_{BH,fructose}\Phi_{BH,fructose}X_{BH}$$

$$\frac{dS_{formate}}{dt} = \alpha_{RI,formate}\Phi_{RI}X_{RI} + \alpha_{FP,formate}\Phi_{FP}X_{FP} - \nu_{BH,formate}\Phi_{BH,formate}X_{BH}$$

$$\frac{dS_{acetate}}{dt} = -\nu_{RI,acetate}\Phi_{RI,acetate}X_{RI} - \nu_{FP,acetate}\Phi_{FP,acetate}X_{FP} + \alpha_{BH,acetate}\Phi_{BH}X_{BH}$$

$$\frac{dS_{butyrate}}{dt} = \alpha_{RI,butyrate}\Phi_{RI}X_{RI} + \alpha_{FP,butyrate}\Phi_{FP}X_{FP}$$

$$\frac{dS_{unknown}}{dt} = -\nu_{FP,unknown}\Phi_{FP}X_{FP}$$

$$\frac{dS_{H_2}}{dt} = \alpha_{RI,H_2}\Phi_{RI}X_{RI} + \alpha_{RI,H_2}\Phi_{BH}X_{BH}$$

$$\frac{dS_{CO_2}}{dt} = \alpha_{RI,CO_2}\Phi_{RI}X_{RI} + \alpha_{FP,CO_2}\Phi_{FP}X_{FP} + \alpha_{BH,CO_2}\Phi_{BH}X_{BH}$$

$$\Phi_{RI,acetate} = \Gamma_{RI}(Q_{RI})\mu_{RI}w_{RI}\frac{S_{fructose}}{K_{RI,fructose}+S_{fructose}}\frac{S_{acetate}}{K_{RI,acetate}+S_{acetate}}$$

$$\Phi_{FP,acetate} = \Gamma_{FP}(Q_{FP})\mu_{FP}w_{FP}\frac{S_{unknown}}{K_{FP,unknown}+S_{unknown}}\frac{S_{acetate}}{K_{FP,fructose}+S_{fructose}}\frac{S_{acetate}}{K_{FP,acetate}+S_{acetate}}$$

$$\Phi_{BH,fructose} = \Gamma_{BH}(Q_{BH})\mu_{BH}\frac{S_{fructose}}{K_{BH,fructose}+S_{fructose}}$$

$$\Phi_{BH,fructose} = \Gamma_{BH}(Q_{BH})\mu_{BH}w_{BH}\frac{S_{formate}}{K_{BH,formate}+S_{formate}}$$

The $\alpha$ and $\nu$ parameters are production and consumption rates, respectively.

Species abundance is measured in $10^8$ bacterial counts/mL, metabolite concentration in mM, the unit of $\mu$ is 1/h, the unit of $K$ is mM, the unit of $\alpha$ and of $\nu$ is mM/($10^8$ bacterial counts/mL) and $\omega$ is dimensionless.

The model assumes that death rates are negligible, that metabolites are produced in proportion to the growth rate of the strains, that metabolites are not transformed in the bioreactor except through the strains themselves and, crucially, that the strains do not alter their metabolism in the presence of interaction partners. Furthermore, Monod kinetics assumes that bacteria grow exponentially at low abundances, that bacterial growth is limited only by the limiting substrate concentration and that the maximal specific growth rate and the Monod constant do not change over time. A number of these assumptions are met by taking the lag phase into account, by omitting data points from the stationary phase and by including the unknown compound for *F. prausnitzii* A2-165 growth.

Carbon dioxide and hydrogen gas consumption by *B. hydrogenotrophica* S5a33 is not included in the final version of the model. We tried to account for carbon dioxide consumption with a multiplicative term in *B. hydrogenotrophica* S5a33's growth rate. However, this did not improve the model fit. As the model without carbon dioxide describes *R. intestinalis* L1-82/*B. hydrogenotrophica* S5a33 biculture dynamics well, we assume that *B. hydrogenotrophica* S5a33 grew mostly heterotrophically on fructose and formate , and that the hydrogen gas and carbon dioxide produced by *R. intestinalis* L1-82 did not reach sufficient concentrations in the head space to allow autotrophic growth.

The model definition is available as *Source code 1* in Python (Model definition).

## Model parameterization

We parameterized our model on monocultures alone (parameterization 1) and then on mono- and bi-cultures (parameterization 2). The model was fitted using the function fmin() from the scipy Python package (*Jones et al., 2001*), to minimize the normalized root mean square error (RMSE). During fitting, the biological replicate(s) of a mono- or bi-culture that gave the best overall fit were selected by trial and error.

An initial estimate of the parameters was obtained by manually fitting the data iteratively. The initial concentration of the unknown compound was set to 30 mM. Samples taken after the end of the log phase, when the bacterial counts started to decline, were omitted from the fitting. Parameterization 2 consisted of several steps, as fitting all parameters at once did not lead to convergence, because of the nonlinear growth rates. First, parameters for *F. prausnitzii* A2-165 were obtained from two *F. prausnitzii* A2-165 monocultures. The consumption parameters of *B. hydrogenotrophica* S5a33 were obtained from *F. prausnitzii* A2-165/*B. hydrogenotrophica* S5a33 bi-cultures with initial acetate; afterwards, the maximal specific growth rates and half-saturation (Monod) constants were obtained from the same bi-cultures. The parameters for *R. intestinalis* L1-82 were obtained from a *R. intestinalis* L1-82/*B. hydrogenotrophica* S5a33 bi-culture with acetate. Lag phases were calculated as the time to reach $\Gamma_i(Q_i) = 0.5$:

$$\text{lag phase} = -\ln(Q_i(0))/\mu_i$$

$Q_i(0)$ was estimated by visual inspection of the log plots. Model parameters obtained and maximal abundances predicted with both parameterizations as well as estimated lag phases and experiment-specific RMSE values are provided in *Supplementary file 2*. Data and model fits were plotted with Python's matplotlib (*Hunter, 2007*).

## RNA extraction and sequencing

Total RNA was extracted from RNAlater-treated samples using the phenol-free total RNA purification kit coupled with DNase I treatment (VWR International) according to the manufacturer's protocol for Gram-positive bacteria. A secondary DNAse digestion was performed using the Ambion TURBO DNA-free DNase Treatment and Removal Reagents Kit (Thermo Fisher Scientific), after which the samples were purified using the RNA Clean and Concentrator−25 kit (Zymo Research, Irvine, CA, USA) according to the manufacturer's instructions.

The eluted RNA was stored at −80°C. The absence of DNA contamination was evaluated using PCR (35 or 40 cycles) and gel electrophoresis. The concentrations of the samples were determined with a Nanodrop, and with a Qubit 2.0 fluorometer using the Qubit dsDNA HS Assay Kit (Thermo Fisher Scientific). RNA integrity, expressed as the RNA integrity number (RIN), and yield were determined using RNA Nano/Pico 6000 LabChips (Agilent Technologies, Santa Clara, CA, USA), which were run in an Agilent 2100 Bioanalyzer (Agilent Technologies). Most of the RINs were above 7, but the RINs of three *B. hydrogenotrophica* S5a33 monoculture samples at 3 hr, 9 hr and 15 hr were around 2.6, and in four cases (*B. hydrogenotrophica* S5a33 monoculture at 15 hr, *F. prausnitzii* A2-165 monoculture at 9 hr, and for both tri-culture replicates at 3 hr), the RINs could not be determined. By pooling over three extraction rounds, however, sufficient RNA for sequencing (minimum of 536 ng and median of 2800 ng) was obtained for all samples.

Library preparation encompassed the use of Ribozero rRNA removal for Gram-positive bacteria and the Illumina TruSeq stranded mRNA Library preparation kit (Illumina, San Diego, CA, USA). Library preparation was performed without the mRNA purification step, according to the manufacturer's instructions. The enriched libraries were sequenced on an Illumina NextSeq 500 instrument (paired-end, 2 × 76 bp reads, Mid output kit, Illumina). From the Illumina platform, paired-end reads in FASTQ format (CASAVA 1.8, Phred + 33) were obtained and separated into distinct files for each single-end read and for each sample.

## RNA-seq analysis

The analysis of the raw sequencing reads was performed as follows: reads were trimmed using Trimmomatic (*Bolger et al., 2014*) with the following parameters 'CROP:74 HEADCROP:10 SLIDING-WINDOW:4:15 MINLEN:51', to remove initial and last bases which had biases in their nucleotide content as reported by FastQC (*Andrews, 2010*), to remove stretches of low-quality bases and to

keep reads with at least 51 bases after trimming. FastQC was re-run on the trimmed data to ensure that the previous biases were corrected. SortMeRNA (*Kopylova et al., 2012*) was used with default parameters and included databases to remove rRNA reads.

With the remaining non-rRNA reads, we ran MetaPhlAn2 (*Truong et al., 2015*) with default parameters and database, and mash screen (*Ondov and Philippy, 2017*; *Ondov et al., 2016*) with default parameters against the complete RefSeq genomes and plasmids database, to search for potential contaminants. Both MetaPhlAn2 and the top hits from mash screen found the correct bacterial genomes from the three strains used in this study, together with reads from yeast (*S. cerevisiae* S288c). In addition, low amounts of the phage PhiX174 were reported by mash screen. To quantify the presence of these potential contaminants in our samples accurately, and to quantify gene expression from the cultured bacteria, we mapped the non-rRNA reads to these five strains using Bowtie2 (*Langmead and Salzberg, 2012*), with default parameters except for '–X 800' to allow for longer insert sizes. The reference genomes used are the following: GCF_000156535.1_ASM15653v1_genomic.fna (*R. intestinalis* L1-82), GCF_000157975.1_ASM15797v1_genomic.fna (*B. hydrogenotrophica* S5a33), GCF_000162015.1_ASM16201v1_genomic.fna (*F. prausnitzii* A2-165), CF_000146045.2_R64_genomic.fna (yeast) and NC_001422.1 (PhiX174). Gene expression was quantified using the htseq-count Python script (*Anders et al., 2015*) (with parameter –a 2 to exclude multimapping reads) for all species using their available *.gff reference annotation files. Given the small size of the PhiX174, we quantified the reads mapping to its entire genome rather than its gene expression.

Differential gene expression analysis of the three cultured strains was performed with DESeq2 (*Love et al., 2014*). To remove the effect of the different bacterial compositions in the tri-culture samples, we extracted the reads from each strain prior to the differential expression analysis and analyzed each strain separately. In the DESeq2 design formula, we included two factors: type of culture (mono- or tri-culture) and time (3 hr, 6 hr and 15 hr). The results of the differential expression analyses were computed using a Wald test of the tri-culture versus the mono-culture samples. For each strain, we extracted the genes whose expression changed significantly (with Benjamini-Hochberg adjusted p-value<0.05) in tri-culture and mapped them to different functional annotations downloaded from the IMG database (*Markowitz et al., 2012*): COG categories, COG numbers and KO numbers. The RNA-seq data-processing code is available on GitHub (*Lloréns-Rico, 2018*; copy archived at https://github.com/elifesciences-publications/syntheticGutCommunity).

## Availability of data and code

RNA-seq results have been deposited in the Short Read Archive under the study identifier SRP136465 (https://www.ncbi.nlm.nih.gov/sra/SRP136465). Fermentation data have been submitted to Dryad (doi:10.5061/dryad.g83f29f). The model definition is provided in *Source code 1* (Model definition). The RNA-seq data processing code is provided on GitHub (*Lloréns-Rico, 2018*; copy archived at https://github.com/elifesciences-publications/syntheticGutCommunity).

## Acknowledgements

The technical assistance of Wim Borremans with the metabolite analyses is gratefully acknowledged. We are also grateful for the help of Gino Vrancken and Thi Thuy Duyen Nguyen, who carried out the $B_{12}$ growth experiment for *F. prausnitzii*. Finally, we thank Frédéric Leroy, Stefan Weckx and Geert Huys for helpful discussions.

## Additional information

### Funding

| Funder | Grant reference number | Author |
| --- | --- | --- |
| Vrije Universiteit Brussel | | Kevin D'hoe |
| Fonds Wetenschappelijk Onderzoek | | Kevin D'hoe<br>Karoline Faust<br>Frédéric Moens<br>Verónica Lloréns-Rico |

| | | |
|---|---|---|
| Interuniversity Institute of Bioinformatics in Brussels | | Stefan Vet |
| Horizon 2020 | AD-GUT project 686271 | Jeroen Raes |

The funders had no role in study design, data collection and interpretation, or the decision to submit the work for publication.

## Author contributions

Kevin D'hoe, Investigation, Methodology, Writing—original draft, Writing—review and editing, Performed the fermentation experiments and metabolite measurements, Performed DNA and RNA extraction and qPCR; Stefan Vet, Formal analysis, Investigation, Visualization, Methodology, Writing—original draft, Writing—review and editing; Karoline Faust, Conceptualization, Supervision, Funding acquisition, Investigation, Methodology, Writing—original draft, Project administration, Writing—review and editing; Frédéric Moens, Supervision, Investigation, Helped to acquire the fermentation data; Gwen Falony, Conceptualization, Supervision, Methodology, Project administration, Writing—review and editing; Didier Gonze, Conceptualization, Formal analysis, Supervision, Visualization, Methodology, Writing—review and editing; Verónica Lloréns-Rico, Software, Investigation, Visualization, Methodology, Writing—original draft, Writing—review and editing; Lendert Gelens, Supervision, Visualization, Writing—review and editing; Jan Danckaert, Supervision, Funding acquisition, Writing—review and editing; Luc De Vuyst, Supervision, Methodology, Writing—review and editing; Jeroen Raes, Supervision, Funding acquisition, Project administration, Writing—review and editing

## Author ORCIDs

Stefan Vet (iD) http://orcid.org/0000-0001-7427-4459
Karoline Faust (iD) http://orcid.org/0000-0001-7129-2803
Didier Gonze (iD) http://orcid.org/0000-0002-9800-2412
Verónica Lloréns-Rico (iD) https://orcid.org/0000-0002-0860-5990
Lendert Gelens (iD) http://orcid.org/0000-0001-7290-9561

## Decision letter and Author response

Decision letter https://doi.org/10.7554/eLife.37090.035
Author response https://doi.org/10.7554/eLife.37090.036

# Additional files

**Supplementary files**

• Supplementary file 1. A summary of all fermentation experiments is provided, including experiment identifiers, carbon recoveries and O/R balances. Fermentation experiments selected for RNA-seq or model parameterization are marked accordingly.
DOI: https://doi.org/10.7554/eLife.37090.023

• Supplementary file 2. The table provides parameter values after model parameterization on mono-cultures only and on mono- and bi-cultures (sheet 1), lag phases estimated from experiments (sheet 2), predicted maximal abundances for the two parameterizations (sheet 3) and experiment-specific RMSE and MAE values (sheet 4).
DOI: https://doi.org/10.7554/eLife.37090.024

• Supplementary file 3. This table lists genes that are significantly up-regulated (green) or downregulated (red) in tri-culture as compared to monoculture for all three strains across all time points.
DOI: https://doi.org/10.7554/eLife.37090.025

• Supplementary file 4. The flow cytometry counts obtained for *Faecalibacterium prausnitzii* A2-165 grown with different concentrations of vitamin $B_{12}$.
DOI: https://doi.org/10.7554/eLife.37090.026

• Supplementary file 5. Sequences for strain-specific primers and probes as well as their specificities.
DOI: https://doi.org/10.7554/eLife.37090.027

• Source code 1. Definition of the kinetic model.
DOI: https://doi.org/10.7554/eLife.37090.028

• Transparent reporting form DOI: https://doi.org/10.7554/eLife.37090.029

## Data availability

RNA-seq results have been deposited to the Short Read Archive under the study identifier SRP136465 (https://www.ncbi.nlm.nih.gov/sra/SRP136465). Fermentation data have been submitted to Dryad (doi:10.5061/dryad.g83f29f). Source data has been provided for Figures 3 to 6.

The following datasets were generated:

| Author(s) | Year | Dataset title | Dataset URL | Database and Identifier |
|---|---|---|---|---|
| D'hoe K, Vet S, Faust K, Moens F, Falony G, Gonze D, Lloreńs-Rico V, Gelens L, Danckaert J, De Vuyst L, Raes J | 2018 | Data from: Integrated culturing, modeling and transcriptomics uncovers emergent behavior in a synthetic gut community | http://dx.doi.org/10.5061/dryad.g83f29f | Dryad Digital Repository, 10.5061/dryad.g83f29f |
| D'hoe K, Vet S, Faust K, Moens F, Falony G, Gonze D, Lloreńs-Rico V, Gelens L, Danckaert J, De Vuyst L, Raes J | 2018 | RNA sequencing of the members of a synthetic gut community | https://www.ncbi.nlm.nih.gov/sra/SRP136465 | NCBI BioProject, SRP136465 |

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
