## [Decision Letter]

Thank you for submitting your article "Integrated culturing, modeling and transcriptomics uncovers emergent behavior in a synthetic gut community" for consideration by *eLife*. Your article has been reviewed by four peer reviewers, one of whom is a member of our Board of Reviewing Editors, and the evaluation has been overseen by Wendy Garrett as the Senior Editor.

The reviewers have discussed the reviews with one another and the Reviewing Editor has drafted this decision to help you prepare a revised submission.

Summary:

D'hoe et al. use batch and continuous culture to model and quantify metabolic cross-feeding by three common human-associated taxa: *B. hydrogenotrophica, F. prausnitzii*, and *R. intestinalis*. The authors first characterized the metabolic activity of *B. hydrogenotrophica*, demonstrating that it was highly versatile and that lactate production varied according to growth substrate. Next, they created replicate sets of mono-, di-, and tricultures, which were used to model community growth according to Monod kinetics. They quantified cross-feeding in mixed communities and showed that complex communities could not be modeled from monocultures, but could be well-modeled with di-culture data. Finally, they use RNA seq data to quantify many additional metabolic changes (e.g. vitamin B_12_ synthesis) in complex communities. The findings represent a fundamental and rather interesting contribution to microbiome sciences given the detailed study of a synthetic gut community and the ever-expanding clinical interest in such communities. The paper's approach and methodology are generally well-executed and noteworthy.

Essential revisions:

1) Although only one strain of each species was used in the model, throughout the manuscript the discussion suggests this could be extrapolated to all strains of the species (and at one point, even genera ("*Blautia* consumes formate produced by *Roseburia* and *Faecalibacterium*")). This was not tested in these experiments, and so this generalization cannot be concluded. Therefore, several parts of the manuscript require careful re-writing to clarify strain heterogeneity. It would be helpful to put the discussion in terms of the genomes of the strains used, and the degree of openness of the genomes of representatives of the species sequenced to date. Likewise, please update Discussion to indicate whether models are expected to be robust across different strains within a species and what might happen if experiments were repeated with a different set of strains of the same species? Would strains that have been (in contrast to the culture collection isolates in this study) maintained by an individual over long periods of time and many generations of bacterial cells interact with one another differently?

2) The major conclusion of the paper is heavily based on the assumption of the existence of a single model that best fits the data. Please discuss the justification for choice of model. This could be done either by citation of literature and discussion of limits of Monod kinetics and impact of any violated model assumptions on conclusions, or alternatively by quantifying the fit of the model to a set of models using alternate variations of Monod kinetics from the literature (see Krumins and Fennell, 2014).

3) Please discuss how to interpret RMSE values in Table 1 and the basis used to deem a model well- or poorly-fitting? As RMSE is sensitive to outliers, please include an additional metric quantifying goodness of fit (such as R^2^ (coefficient of determination) or MAE (mean absolute error)) that is less sensitive to outliers.

4) The manuscript shows that *F. prausnitzii* is growth-limited by a heat-labile cofactor, possibly B_12_, and in tri-culture, FP's B_12_biosynthesis genes are downregulated. If time permits, it would improve the impact of the manuscript to show that FP is growth-limited by B_12_ specifically, such as with a simple batch experiment +/- B_12_ supplementation through a sterile filter.

[Editors' note: further revisions were requested prior to acceptance, as described below.]

Thank you for resubmitting your work entitled "Integrated culturing, modeling and transcriptomics uncovers emergent behavior in a synthetic gut community" for further consideration at *eLife*. Your revised article has been favorably evaluated by Wendy Garrett as the Senior Editor and the Reviewing Editor.

The manuscript has been improved. At this juncture, we have advice and suggestions that we think will improve the readability of your manuscript and clarity of the Abstract and Introduction. Please respond to these remaining issues that need to be addressed before acceptance, as outlined below:

As per *eLife* editorial guidelines, a study's title and Abstract should clearly indicate the model system being used for a study. Therefore:

1) Please revise your study title to make it more informative. I am including a suggestion but your suggestions are welcome.

Suggestion: Integrated fermentation culturing, modeling and transcriptomics uncovers complex interactions and emergent behavior in a three-species synthetic human gut community.

2) Please update your Abstract to indicate which three strains are included in your model community. Consider also updating Abstract to give a few key examples of the kinds of taxa behavior that are described.

3) Reviewers previously requested that authors shorten the end of Introduction in particular and authors have clearly made an effort to do so, but further reducing length of last three paragraphs by ~25%, particularly by trimming methods further, and re-ordering a little may improve readability of manuscript.

Here's a proposed revision as a suggestion for a possible way to shorten that text:

"In this study, we created a synthetic community composed of three abundant and typical members of the human gut microbiome: *Faecalibacteriumprausnitzii* A2-165 (Duncan, Hold et al., 2002b), *Roseburiaintestinalis* L1-82 (Duncan, Hold et al., 2002a) and *Blautiahydrogenotrophica* S5a33 (Bernalier, Willems et al., 1996). All three strains were isolated from human feces. They have well-characterized metabolism, draft genomes available, and described potential for cross-feeding. Furthermore, they are of particular medical relevance due to the ability of two of three strains (*R. intestinalis* L1-82 and *F. prausnitzii* A2-165) to produce butyrate, an beneficial short chain fatty acid that is an important energy source for gut epithelial cells (Geirnaert, Calatayud et al., 2016, Rivière et al., 2016). Butyrate producers are often depleted in dysbiotic gut microbiota relative to healthy controls (Antharam, Li et al., 2013, Rivera-Chávez, Zhang et al., 2016). Thus, high butyrate production will likely be a quality criterion for bacterial cocktails designed for therapeutic purposes.

In *R. intestinalis* L1-82, fermentation of carbohydrates results in production of butyrate as well as hydrogen gas and carbon dioxide (Duncan et al., 2002a, Falony, Verschaeren et al., 2009c), whereas *F. prausnitzii* A2-165 produces formate in addition to butyrate and requires acetate for growth (Duncan et al., 2002b, Moens et al., 2016). *B. hydrogenotrophica* S5a33 is able to grow on carbon dioxide and hydrogen gas, but also on glucose and fructose, in all cases generating acetate (Bernalier et al., 1996). Therefore, as Figure 2 illustrates, our community contains multiple cross-feeding and competitive interactions. For instance, all three strains compete for fructose. *B. hydrogenotrophica* S5a33 can use the hydrogen gas generated by *R. intestinalis* L1-82 as well as of the carbon dioxide and formate produced by both *R. intestinalis* L1-82 and *F. prausnitzii* A2-165. In turn, *B. hydrogenotrophica* S5a33 provides acetate that is beneficial to *R. intestinalis* L1-82 and *F. prausnitzii* A2-165. This system thus constitutes a rare example of two strain pairs that simultaneously compete and mutually cross-feed.

To reach our objectives, we created a synthetic model community comprised of three common gut commensal bacteria: *Faecalibacterumprausnitzii, Blautiahydrogenotrophica*, and *Roseburiaintestinalis*. These strains were grown as mono-, di-, or tricultures in 2-L laboratory fermentors in batch mode. We measured growth kinetic parameters, and quantified the dynamics of each combination. We quantified bacteria (flow cytometry, qPCR, OD_600_), growth substrates, and the short chain fatty acids and gases produced by fermentation. Finally, we sequenced the total RNA in selected samples. Figure 1 summarizes our approach. To our knowledge, this is the first study that investigates a synthetic gut community with a combination of mono- and co-cultures, mechanistic modeling and gene expression analysis."

---

## [Author Response]

Essential revisions:1) Although only one strain of each species was used in the model, throughout the manuscript the discussion suggests this could be extrapolated to all strains of the species (and at one point, even genera ("Blautia consumes formate produced by Roseburia and Faecalibacterium")). This was not tested in these experiments, and so this generalization cannot be concluded. Therefore, several parts of the manuscript require careful re-writing to clarify strain heterogeneity. It would be helpful to put the discussion in terms of the genomes of the strains used, and the degree of openness of the genomes of representatives of the species sequenced to date. Likewise, please update Discussion to indicate whether models are expected to be robust across different strains within a species and what might happen if experiments were repeated with a different set of strains of the same species? Would strains that have been (in contrast to the culture collection isolates in this study) maintained by an individual over long periods of time and many generations of bacterial cells interact with one another differently?

We excuse for giving the impression that we are trying to generalize our conclusions beyond the strains discussed. This was not our intention. We have now modified the text as well as the figure captions and the supplementary tables to mention strain instead of species names.

For each of the three strains, there are only few related strains known.

Butyrate production and acetate consumption has been reported for all five *Roseburiaintestinalis* strains (L1-81, L1-82, L1-93, L1-952, L1-8151) originally described by Duncan et al. (2002b). The pyruvate ferredoxin oxidoreductase, which forms a crucial step in this pathway, is present in two further *R. intestinalis* strains (M50/1 and XB6B4).

Concerning *Faecalibacteriumprausnitzii*, both strains originally described by Duncan at al. (2002b) are reported to produce formate and butyrate and to consume acetate. The key enzyme of formate production, formate acetyltransferase, is present in three further *F. prausnitzii* strains (SL3/3, KLE1255 and M21/2). These three strains are also predicted to produce formate in the AGORA database (Magnusdottir et al., 2017).

Bernalier and colleagues originally described two *Blautiahydrogenotrophica* strains, both of which were able to grow autotrophically on H_2_ and CO_2_ (Bernalier et al., 1996). Rey and colleagues worked with the second *B. hydrogenotrophica* strain (S5a36), butdid not demonstrate hydrogen gas or formate consumption directly. However, they showed that in gnotobiotic mice, *B. hydrogenotrophica* S5a36grew betterin the presence than in the absence of *Bacteroidesthetaiotaomicron* and that its Wood-Ljungdahl pathway genes were strongly expressed (Rey et al., 2010).

Concerning modeling: model predictions depend on parameter values, including Monod constants and metabolite consumption and uptake rates, which are hard to obtain from the genome. Without further measurements, it is therefore premature to claim that the dynamics reported here would be the same with another strain combination.

In brief, while we have some indications that the cross-feeding interactions may be preserved across strains within the species considered, we cannot ascertain this without further experiments. We now state this in the Discussion as follows:

“While only one strain for each of the three colonic species considered was tested, the relationships described here likely generalize to species level. […] However, further experiments are required to test whether the potential for cross-feeding is realized in other strain combinations.”

The last question is about the impact of evolution on the interactions. Gut bacteria are known to evolve rapidly in mice (Lourenço et al., 2016) and bacteria isolated from pools around beech tree roots have been reported to develop new cross-feeding interactions within weeks when grown in a community (Lawrence et al., 2012). The interactions reported here involve pathways from the strains' core metabolism, which is less likely to change rapidly. However, since long-term evolution is hard to predict, there is no guarantee that interactions will be preserved when strains are maintained for long periods of time, even if the environment is kept constant.

2) The major conclusion of the paper is heavily based on the assumption of the existence of a single model that best fits the data. Please discuss the justification for choice of model. This could be done either by citation of literature and discussion of limits of Monod kinetics and impact of any violated model assumptions on conclusions, or alternatively by quantifying the fit of the model to a set of models using alternate variations of Monod kinetics from the literature (see Krumins and Fennell, 2014).

First of all, we would like to point out that the major conclusion of our work, i.e. the change of metabolic behavior in the presence of an interaction partner, is not only supported by the model, but also independently by the transcriptomics data.

Like all modeling approaches, our model makes a number of assumptions: that the death rates are negligible, that metabolites are produced proportionally to the abundance of the strains, that metabolites are not transformed in the bioreactor except through the strains themselves and, crucially, that the strains do not alter their metabolism in the presence of interaction partners. More specifically, the Monod model assumes that bacteria grow exponentially at low abundances, that bacterial growth is limited only by the limiting substrate concentration and that the maximal specific growth rate and the Monod constant do not change over time. The growth of our strains is characterized by a lag phase, which is explicitly accounted for in our model. To estimate Monod parameter values, we omitted the first data points (lag phase) as well as the last data points (bacterial decline due to death). We also included additional substrates when necessary. While we cannot ascertain that all assumptions are fulfilled in our system, the fact that we obtained a reasonable fit suggests that there are no large deviations from these assumptions.

Concerning the alternative models discussed by Krumins and Fennell: as stated by these authors themselves, zero- and first-order models are special cases of the Monod model, which are applicable when the substrate concentration S is much larger than the Monod constant K or much smaller, respectively. In the zero-order case, the growth rate does not depend on the nutrient concentration and the substrate decreases linearly, which does not fit the shape of our fructose consumption curves. As for the first-order case, it was not selected by our fitting procedure, since after parameterization, substrate concentrations and Monod constants are of the same order of magnitude. Thus, the conditions justifying the special cases are not fulfilled. However, to further investigate the first-order case, we simplified our equations to impose first-order kinetics and repeated the bi-culture fit (parameterization 2, with the *R. intestinalis* L1-82 mono-culture instead of the *R. inrestinalis* L1-82/*B. hydrogenotrophica* S5a33 bi-culture, to avoid negative parameter values). The first-order model did not fit the co-culture data as well as the standard Monod model according to the RMSE values (mean for the standard Monod fit: 0.47, mean for the first-order fit: 0.62, paired, two-sided Wilcoxon test p-value: 0.0003). In addition, with the first-order model, it is no longer possible to distinguish the impact of fructose and the unknown co-factor on the growth of *F. prausnitzii* A2-165.

Krumins and Fennell also discuss the Best model, which was developed to take mass transfer limitations into account. This model comes with an additional parameter, the exchange constant k. Since we have no indication that mass transfer limitation plays a role in our system, there is no reason to use this more complex model.

In summary, we chose the simplest model that encoded our biological knowledge and was able to describe the observed dynamics. We have neither a biological motivation for a more complex kinetic model, nor a practical reason, since the goodness of fit of the simple model was sufficient for our purpose. Complications such as Hill functions, exchange constants and time-dependent Monod constants only increase the risk of over-fitting without gain.

However, as we stated in the Discussion, we do see the need for a metabolic model that includes the internal metabolism, since it can potentially account for metabolic changes in response to interaction partners. The development of such a model for our system is a topic of future research.

In response to this comment, we now discuss better the assumptions of our model in the section on model definition:

“The model assumes that death rates are negligible, that metabolites are produced proportionally to the growth rate of the strains, that metabolites are not transformed in the bioreactor except through the strains themselves and, crucially, that the strains do not alter their metabolism in the presence of interaction partners. […] A number of these assumptions are met by taking the lag phase into account, by omitting data points from the stationary phase, and by including the unknown compound for *F. prausnitzii* A2-165.”

3) Please discuss how to interpret RMSE values in Table 1 and the basis used to deem a model well- or poorly-fitting? As RMSE is sensitive to outliers, please include an additional metric quantifying goodness of fit (such as R^2^ (coefficient of determination) or MAE (mean absolute error)) that is less sensitive to outliers.

The RMSE values of the first parameterization are larger for co-cultures than for mono-cultures, implying that the fit to co-cultures was less accurate than the fit to mono-cultures. For the second parameterization, the RMSE values show the opposite trend. This means that the model cannot fit both mono- and co-cultures equally well, which we think is due to metabolic changes in response to interaction partners. We can derive this conclusion without establishing a threshold RMSE value, beyond which a fit is deemed bad.

The RMSE, MAE and R^2^ are global measures, which average the goodness of fit over the abundances and concentrations of all strains and nutrients considered. In general, we think that such global measures are of limited value when it comes to assessing the goodness of fit to non-linear functions, since a small shift of the fit with respect to the original function can result in huge deviations. We therefore assessed goodness of fit primarily visually. For this reason, we have included figures visualizing the fit to each experiment for both parameterizations in the main and in the supplement.

We have now added the MAE to Supplementary file 2. The MAE values are almost perfectly correlated to the RMSE values (0.99 for both parameterizations). The R^2^, being based on (scale-invariant) correlation, cannot capture absolute differences and we therefore do not include it here.

4) The manuscript shows that F. prausnitzii is growth-limited by a heat-labile cofactor, possibly B_12_, and in tri-culture, FP's B_12_biosynthesis genes are downregulated. If time permits, it would improve the impact of the manuscript to show that FP is growth-limited by B_12_ specifically, such as with a simple batch experiment +/- B_12_ supplementation through a sterile filter.

Following this suggestion, we carried out a growth experiment in test tubes, with three different concentrations of filter-sterilized B_12_ solution as well as a control without added B_12_, each in three biological replicates. *F. prausnitzii* abundance was quantified via flow cytometry. We did not see a significant difference in abundance at 48 h and a significant negative effect on growth at 24 h (new Supplementary file 4). We assume that *F. prausnitzii* is more strongly growth-limited by another of its required co-factors than by vitamin B_12_. We currently have no adequate explanation for the negative effect of B_12_ at 24 h.

[Editors' note: further revisions were requested prior to acceptance, as described below.]

Thank you for resubmitting your work entitled "Integrated culturing, modeling and transcriptomics uncovers emergent behavior in a synthetic gut community" for further consideration at eLife. Your revised article has been favorably evaluated by Wendy Garrett as the Senior Editor and the Reviewing Editor.The manuscript has been improved. At this juncture, we have advice and suggestions that we think will improve the readability of your manuscript and clarity of the Abstract and Introduction. Please respond to these remaining issues that need to be addressed before acceptance, as outlined below:As per eLife editorial guidelines, a study's title and Abstract should clearly indicate the model system being used for a study. Therefore:1) Please revise your study title to make it more informative. I am including a suggestion but your suggestions are welcome.Suggestion: Integrated fermentation culturing, modeling and transcriptomics uncovers complex interactions and emergent behavior in a three-species synthetic human gut community.2) Please update your Abstract to indicate which three strains are included in your model community. Consider also updating Abstract to give a few key examples of the kinds of taxa behavior that are described.3) Reviewers previously requested that authors shorten the end of Introduction in particular and authors have clearly made an effort to do so, but further reducing length of last three paragraphs by ~25%, particularly by trimming methods further, and re-ordering a little may improve readability of manuscript.Here's a proposed revision as a suggestion for a possible way to shorten that text:"In this study, we created a synthetic community composed of three abundant and typical members of the human gut microbiome: Faecalibacterium prausnitzii A2-165 (Duncan, Hold et al., 2002b), Roseburia intestinalis L1-82 (Duncan, Hold et al., 2002a) and Blautia hydrogenotrophica S5a33 (Bernalier, Willems et al., 1996). All three strains were isolated from human feces. They have well-characterized metabolism, draft genomes available, and described potential for cross-feeding. Furthermore, they are of particular medical relevance due to the ability of two of three strains (R. intestinalis L1-82 and F. prausnitzii A2-165) to produce butyrate, an beneficial short chain fatty acid that is an important energy source for gut epithelial cells (Geirnaert, Calatayud et al., 2016, Rivière et al., 2016). Butyrate producers are often depleted in dysbiotic gut microbiota relative to healthy controls (Antharam, Li et al., 2013, Rivera-Chávez, Zhang et al., 2016). Thus, high butyrate production will likely be a quality criterion for bacterial cocktails designed for therapeutic purposes.In R. intestinalis L1-82, fermentation of carbohydrates results in production of butyrate as well as hydrogen gas and carbon dioxide (Duncan et al., 2002a, Falony, Verschaeren et al., 2009c), whereas F. prausnitzii A2-165 produces formate in addition to butyrate and requires acetate for growth (Duncan et al., 2002b, Moens et al., 2016). B. hydrogenotrophica S5a33 is able to grow on carbon dioxide and hydrogen gas, but also on glucose and fructose, in all cases generating acetate (Bernalier et al., 1996). Therefore, as Figure 2 illustrates, our community contains multiple cross-feeding and competitive interactions. For instance, all three strains compete for fructose. B. hydrogenotrophica S5a33 can use the hydrogen gas generated by R. intestinalis L1-82 as well as of the carbon dioxide and formate produced by both R. intestinalis L1-82 and F. prausnitzii A2-165. In turn, B. hydrogenotrophica S5a33 provides acetate that is beneficial to R. intestinalis L1-82 and F. prausnitzii A2-165. This system thus constitutes a rare example of two strain pairs that simultaneously compete and mutually cross-feed.To reach our objectives, we created a synthetic model community comprised of three common gut commensal bacteria: Faecalibacterum prausnitzii, Blautia hydrogenotrophica, and Roseburia intestinalis. These strains were grown as mono-, di-, or tricultures in 2-L laboratory fermentors in batch mode. We measured growth kinetic parameters, and quantified the dynamics of each combination. We quantified bacteria (flow cytometry, qPCR, OD_600_), growth substrates, and the short chain fatty acids and gases produced by fermentation. Finally, we sequenced the total RNA in selected samples. Figure 1 summarizes our approach. To our knowledge, this is the first study that investigates a synthetic gut community with a combination of mono- and co-cultures, mechanistic modeling and gene expression analysis."

We are glad to learn that our revised article has been favorably evaluated. We

have addressed the remaining issues and also provided the missing source data

files.

References

Krumins V & Fennel D. (2014). Identifying the Correct Biotransformation Model from Polychlorinated Biphenyl and Dioxin Dechlorination Batch Studies. Environmental Engineering Science, 31(10). doi: 10.1089/ees.2013.0463

Lawrence D, Fiegna F, Behrends V, Bindy JG, Phillimore AB, Bell T & Barraclough TG. (2012). Species Interactions Alter Evolutionary Responses to a Novel Environment. Plos Biology. doi: 10.1371/journal.pbio.1001330

Lourenço M, Ramiro R, Güleresi D, Barroso-batista J, Xavier KB, Gordo I & Sousa S. (2016). A Mutational Hotspot and Strong Selection Contribute to the Order of Mutations Selected for during *Escherichia coli* Adaptation to the Gut. PLoS Genetics. doi: 10.1371/journal.pgen.1006420